# TRIM5α restricts poxviruses and is antagonized by CypA and the viral protein C6

Yiqi Zhao[1,2,6], Yongxu Lu[1,2,6], Samuel Richardson[3], Meghna Sreekumar[1], Jonas D. Albarnaz[1,5 ✉] & Geoffrey L. Smith[1,2,3,4 ✉]

Human tripartite motif protein 5α (TRIM5α) is a well-characterized restriction factor for some RNA viruses, including HIV[1–5]; however, reports are limited for DNA viruses[6,7]. Here we demonstrate that TRIM5α also restricts orthopoxviruses and, via its SPRY domain, binds to the orthopoxvirus capsid protein L3 to diminish virus replication and activate innate immunity. In response, several orthopoxviruses, including vaccinia, rabbitpox, cowpox, monkeypox, camelpox and variola viruses, deploy countermeasures. First, the protein C6 binds to TRIM5 via the RING domain to induce its proteasome-dependent degradation. Second, cyclophilin A (CypA) is recruited via interaction with the capsid protein L3 to virus factories and virions to antagonize TRIM5α; this interaction is prevented by cyclosporine A (CsA) and the non-immunosuppressive derivatives alisporivir and NIM811. Both the proviral effect of CypA and the antiviral effect of CsA are dependent on TRIM5α. CsA, alisporivir and NIM811 have antiviral activity against orthopoxviruses, and because these drugs target a cellular protein, CypA, the emergence of viral drug resistance is difficult. These results warrant testing of CsA derivatives against orthopoxviruses, including monkeypox and variola.

Vaccinia virus (VACV) is the live vaccine used to eradicate smallpox and is currently being used to immunize at-risk populations against monkeypox virus (MPXV), the cause of the disease mpox. VACV, cowpox virus (CPXV), MPXV, camelpox virus (CMLV) and variola virus (VARV), the cause of smallpox, are all orthopoxviruses and are immunologically cross-protective. After the eradication of smallpox, VACV has continued to be studied as a platform for vaccine development, as an oncolytic agent and as an excellent model for studying virus–host interactions. In particular, VACV and other orthopoxviruses encode scores of proteins that subvert the host immune system[8,9].

Previously, a proteomic study had revealed that infection of human fibroblasts with VACV strain Western Reserve (VACV-WR) induced the reduction of 265 cellular proteins (approximately 3% of those quantified), mostly by proteasome-mediated degradation[10]. To explain this targeted degradation, one hypothesis is that the degraded proteins have antiviral activity and so are eliminated as a viral evasion strategy. For histone deacetylase 4 (HDAC4) and HDAC5, this hypothesis was shown to be correct, and HDAC4 was antiviral and functions in the type I interferon (IFN)-mediated signal transduction pathway to induce expression of IFN-stimulated genes[11]. Another protein, TRIM5, was also degraded during VACV infection and is the subject of this study.

## The VACV protein C6 binds to the TRIM5α RING domain

The proteasomal degradation of TRIM5α observed in VACV-infected TERT-immortalized human fetal foreskin fibroblasts by mass spectrometry[10] (Fig. 1a) was confirmed by immunoblotting in HeLa cells (Fig. 1b). Only the TRIM5α isoform was detected in these cells. To identify the VACV protein (or proteins) responsible, VACV mutants lacking blocks of genes from near either genomic terminus[12] were utilized and showed that the mutant v6/2 was unable to degrade TRIM5α (Fig. 1c). Bioinformatic analysis of the proteins encoded by the genes missing in v6/2 (ref. 13) and analysis of single-gene deletion mutants identified the gene *C6L* as being necessary for TRIM5α degradation (Fig. 1d). C6 is a multifunctional antagonist of innate immunity that is expressed early during infection and has a predicted BCL-2-like fold[10,11,14–16]. A cell line engineered to express the VACV-WR protein C6 also reduced TRIM5α, showing that no other viral protein is needed (Fig. 1e). TRIM5α is an E3 ubiquitin ligase and can regulate its own stability via autoubiquitylation[17–19]. To determine whether C6-mediated TRIM5α degradation required TRIM5α E3 ligase activity, mutant TRIM5α(N70A) lacking autoubiquitylation activity[18] was examined and found to be degraded as for the wild-type (WT) protein (Fig. 1f and Extended Data Fig. 1a). Thus, degradation of TRIM5α is probably mediated by one or more unknown cellular E3 ligases that C6 might co-opt to ubiquitylate TRIM5α, leading to proteasomal degradation.

Infection of HEK293T cells with VACV strains expressing TAP-tagged C6 or N1, another BCL-2-like VACV immunomodulatory protein[20], followed by affinity purification demonstrated that C6 co-precipitates with TRIM5α when expressed during infection at endogenous levels (Fig. 1g). Ectopic expression of epitope-tagged viral proteins confirmed that C6, but not the VACV protein B14 (a BCL-2-like VACV protein that

[1]Department of Pathology, University of Cambridge, Cambridge, UK. [2]Sir William Dunn School of Pathology, University of Oxford, Oxford, UK. [3]The Pirbright Institute, Surrey, UK. [4]Chinese Academy of Medical Sciences–Oxford Institute, University of Oxford, Oxford, UK. [5]Present address: Cambridge Institute for Medical Research, University of Cambridge, Cambridge, UK. [6]These authors contributed equally: Yiqi Zhao, Yongxu Lu. ✉e-mail: jd732@cam.ac.uk; geoffrey.smith@path.ox.ac.uk

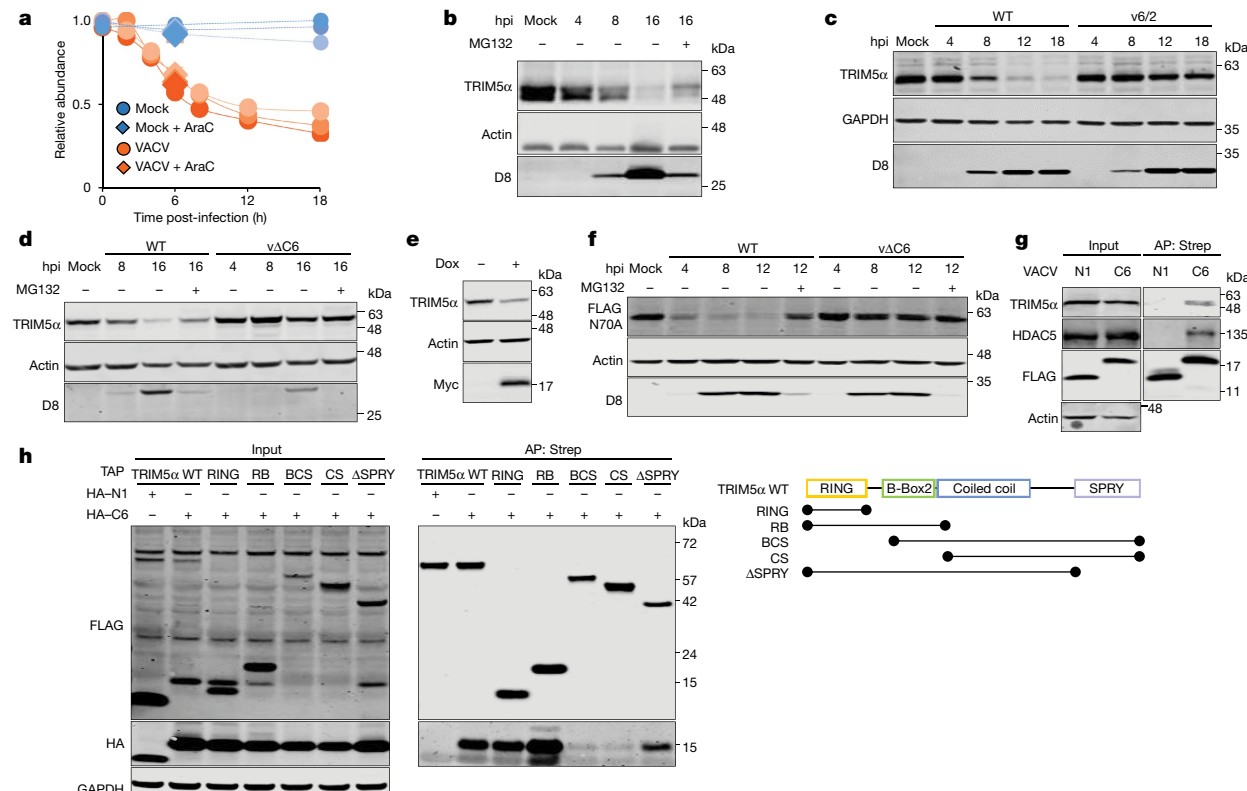

**Fig. 1 | The VACV protein C6 induces proteasomal degradation of TRIM5α.**
**a**, Temporal abundance of TRIM5 during VACV-WR infection of TERT-immortalized human fetal foreskin fibroblast (HFFF-TERT) cells measured by mass spectrometry. *n* = 3 per condition (from ref. 10). Cytosine arabinoside (AraC) was added where indicated. **b–f**, Immunoblots showing TRIM5α abundance in: VACV-infected HeLa cells at the indicated times post-infection with (+) or without (−) MG132 (**b**); HeLa cells after infection with VACV or mutant v6/2 lacking genes near the left genomic terminus (**c**); HFFF-TERT cells after infection with VACV-WR WT or mutant vΔC6 lacking the gene *C6L* (**d**); HEK293T cells inducibly expressing Myc-tagged C6 (+150 ng ml⁻¹ doxycycline (Dox) for 24 h) (**e**); and T-REx-293 *TRIM5⁻/⁻* cells complemented with inducible expression of the TAP-tagged TRIM5α mutant N70A and infected with WT VACV-WR or

vΔC6 (**f**). **g**, Endogenous C6 and TRIM5α co-precipitate during VACV infection. HEK293T cells were infected with VACV-expressing TAP-tagged C6 or N1. HDAC5 was used as a positive control for C6 co-precipitation (unpublished data). **h**, Mapping of the TRIM5α domains required for C6 interaction. T-REx-293 *TRIM5⁻/⁻* cells were co-transfected with HA-tagged C6 or N1 and TAP-tagged TRIM5α WT and mutants lacking domains, which are indicated in the right schematics. In **b**,**d**,**f**, MG132 was added at 2 h post-infection (hpi) (+). In **g**,**h**, TAP-tagged proteins were precipitated using Strep-Tactin beads. In **b–h**, inputs and affinity-purified (AP) proteins or protein extracts were analysed by SDS–PAGE and immunoblotted for the indicated epitope or protein. GAPDH, α-actin and D8 were controls for equal loading and viral infection, respectively. Data from **b–h** are representative of three independent experiments.

antagonizes NF-κB[14,21]), co-precipitated with endogenous TRIM5α (Extended Data Fig. 1b). HDAC5 and IKKβ were analysed as known binding partners of C6 and B14, respectively[21]. Use of recombinant protein produced in vitro by the wheat germ transcription and translation system showed that the interaction of C6 and TRIM5α was not mediated via other mammalian proteins and so probably is direct (Extended Data Fig. 1c). By expressing several tagged TRIM5α mutants that lack different domains, C6 was shown to interact with the TRIM5α N-terminal RING domain (Fig. 1h). Finally, C6 enhances ubiquitylation of TRIM5α in the presence of ectopic ubiquitin or a ubiquitin mutant containing only K63, but not only K48, indicating that K63 ubiquitylation mediates C6-induced degradation of TRIM5α (Extended Data Fig. 1d).

## TRIM5α restricts VACV

Next, the antiviral activity of TRIM5 was investigated in T-REx-293 and HeLa cells engineered, by CRISPR–Cas9-mediated genome editing, to lack all TRIM5 isoforms. Loss of TRIM5 was confirmed in two clones for each cell type by immunoblotting (Extended Data Fig. 2a,b) and DNA sequencing. The replication and spread of VACV in these cells were investigated using WT or an eGFP-tagged VACV-WR strain[22]. In both T-REx-293 (Fig. 2a) and HeLa (Extended Data Fig. 2c) cells lacking TRIM5, the yields of infectious virus after high multiplicity of infection (MOI)

were enhanced. Similar data on virus titre and size of virus plaques were obtained after low MOI (Fig. 2b and Extended Data Fig. 2d–f). Although TRIM5α is known to be antiviral for some RNA viruses, other isoforms such as TRIM5γ and TRIM5δ are proviral, by antagonizing TRIM5α[23]. To test the roles of these isoforms against VACV, each isoform was overexpressed in WT T-REx-293 cells and the replication and spread of VACV were examined. Whereas overexpression of TRIM5α was antiviral, the expression of TRIM5γ or TRIM5δ was proviral (Extended Data Fig. 2g–k), due to dominant negative antagonism of TRIM5α in WT cells. Next, the *TRIM5⁻/⁻* T-REx-293 cells were complemented by stable transfection to express TRIM5α, TRIM5γ or TRIM5δ inducibly (Extended Data Fig. 2l), and the growth and spread of VACV in these cells were investigated. This showed that expression of TRIM5α, but not TRIM5γ or TRIM5δ, reduced virus plaque size and yields of infectious virus (Fig. 2c,d and Extended Data Fig. 2m,n).

To address how TRIM5α was antiviral for VACV, the *TRIM5⁻/⁻* T-REx-293 cells were complemented by inducible expression of the TRIM5α mutants L19R, N70A, R119E and one lacking the SPRY domain (also known as PRY-SPRY or B30.2) (ΔSPRY) (Extended Data Fig. 2o). L19R is catalytically defective, strongly impaired in its ability to synthesize anchored and unanchored polyubiquitin chains, as well as monoubiquitylation[18]. N70A has impaired monoubiquitylation and anchored polyubiquitin chain synthesis, but has retained its ability to synthesize

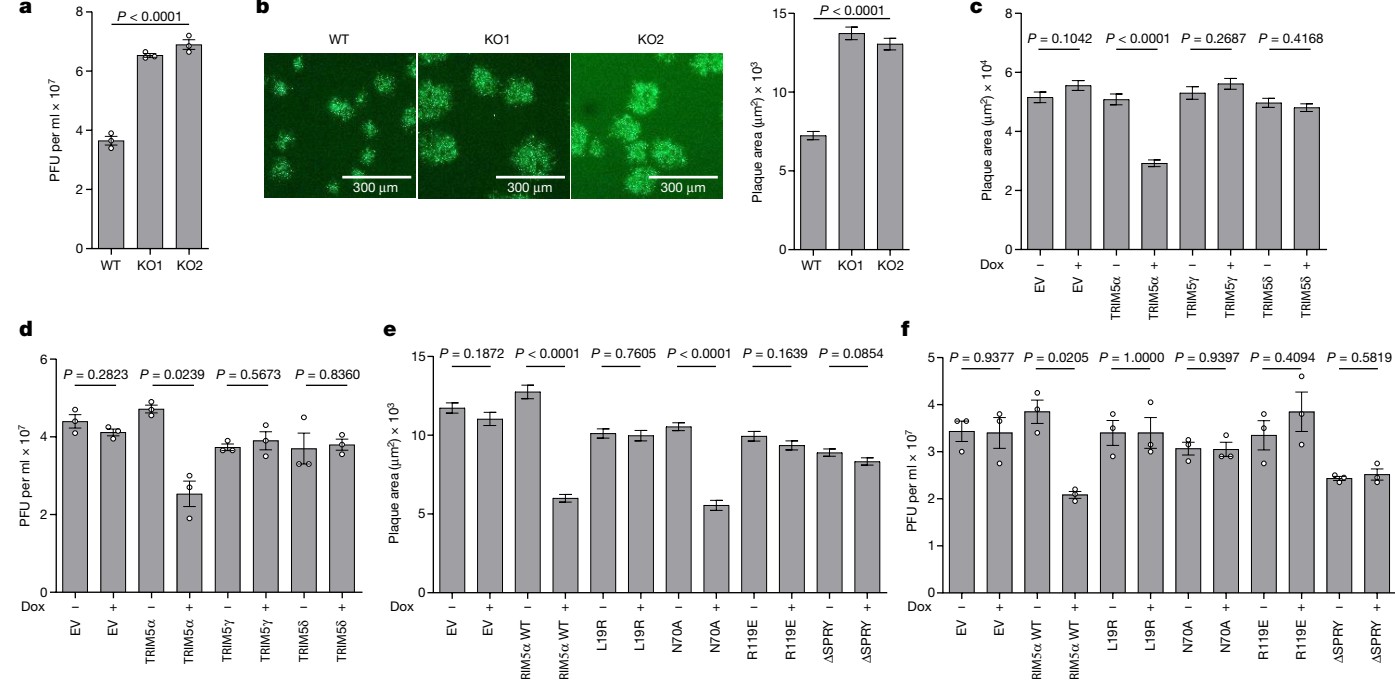

**Fig. 2 | TRIM5α is a VACV restriction factor. a**, Infectious VACV titres following infection of T-REx-293 (WT) and derivative *TRIM5*[−/−] (KO1 or KO2) cells with VACV-WR at 5 plaque-forming units (PFU) per cell for 16 h. $n = 3$ per condition. **b**, Plaque image (left) and plaque area quantification (right) 24 hpi for the cells described in **a** with VACV A5–GFP. $n \geq 114$ per condition. **c**, Plaque area quantification 24 hpi with VACV A5–GFP of T-REx-293 *TRIM5*[−/−] cells engineered to inducibly express (+Dox) TAP-tagged TRIM5α, TRIM5γ and TRIM5δ isoforms. $n \geq 87$ per condition. EV, empty vector. **d**, Infectious VACV titres following infection of cells described in **c** at 5 PFU per cell for 16 h. $n = 3$ per condition. **e**, Plaque area quantification 24 hpi with VACV A5–GFP of T-REx-293 *TRIM5*[−/−] cells engineered to inducibly express (+Dox) the following TAP-tagged TRIM5α mutants: L19R, N70A, R119E and ΔSPRY. $n \geq 54$ per condition. **f**, Infectious VACV titres following infection of cells described in **e** at 5 PFU per cell for 16 h. $n = 3$ per condition. In **c**,**d**, cells were treated with 150 ng ml[−1] doxycycline from seeding for 24 h to express tagged proteins before harvest or infection. In **e**,**f**, doxycycline was added at 24 h after seeding. Data shown in **b**,**c**,**e** are representative of three independent experiments, and in **a**,**d**,**f** are from two independent experiments. Data from **a**,**b** were analysed using one-way Welch's analysis of variance (ANOVA) test. Data from **c**–**f** were analysed using two-tailed unpaired Student's *t*-test. Analyses were performed on GraphPad Prism. Data are mean ± s.e.m.

free polyubiquitin chains, which subsequently activate innate immune signalling pathways such as NF-κB and AP1 (ref. 18). R119E is defective in oligomerization[24] and the SPRY domain is required for retrovirus capsid recognition[25]. Whereas expression of TRIM5α WT and N70A were antiviral after low MOI and reduced plaque size, the mutants L19R, R119E and ΔSPRY were not (Fig. 2e and Extended Data Fig. 2p,q), indicating that polyubiquitylation, oligomerization and SPRY domains were all needed for antiviral activity during low MOI. After high MOI, the yields of VACV showed a similar sensitivity to the TRIM5α mutants, except for N70A, which was no longer antiviral (Fig. 2f). Thus, high MOI can overcome the antiviral effect of TRIM5α(N70A), suggesting that this might reflect viral evasion from the innate immune response to infection (see below), rather than inhibition of virus replication itself. This implies that TRIM5α-mediated restriction may operate by influencing both virus replication and innate immunity.

## CypA is proviral but TRIM5α dependent

In retroviruses, the antiviral activity of TRIM5α is negated by the proviral activity of CypA (encoded by *PPIA*, also known as *CypA*) that binds to the same viral capsid protein and antagonizes TRIM5α binding[4,5]. Previous studies have reported that CypA is also recruited to the VACV core[26] and that CsA and non-immunosuppressive derivatives are antiviral[27,28]. To investigate this further, cell lines lacking either CypA (Extended Data Fig. 3a) or both CypA and TRIM5α (Extended Data Fig. 3b) were constructed. In cells lacking only CypA, VACV plaque size and viral titres were reduced (Fig. 3a–c and Extended Data Fig. 3c), whereas in cells also lacking TRIM5α, loss of CypA had no effect (Fig. 3a,d,e and Extended Data Fig. 3d). Thus, CypA was only proviral in the presence of TRIM5α.

In addition, although increasing doses of CsA diminished VACV plaque size in WT cells, in the absence of TRIM5α, the drug was not antiviral (Fig. 3f and Extended Data Fig. 3e). Thus, both the proviral activity of CypA and the antiviral activity of CsA are dependent on TRIM5α.

CypA is a prolyl isomerase and can aid protein folding but, when bound by CsA, can form an immunosuppressive complex that affects the phosphatase activity of calcineurin and thereby the transcriptional activity of NFAT[29–31]. To investigate which activity was needed for pro-poxviral potency, either WT CypA or the catalytically defective CypA mutants R55A and F113A were expressed inducibly in the *CypA*[−/−] T-REx-293 cells (Extended Data Fig. 3f) and the plaque size and yield of infectious virus were measured. Whereas the expression of WT CypA was proviral, the mutants were not (Fig. 3g,h and Extended Data Fig. 3g,h) and thus the enzymatic activity of CypA was needed for proviral activity.

## CypA and TRIM5α bind to the capsid protein L3

In retroviruses, both CypA and TRIM5α bind to the same capsid protein, and which retroviral capsids can be bound is determined by variation in the TRIM5α SPRY domain[32–34]. To investigate which VACV protein (or proteins) TRIM5α and CypA bind to, an unbiased proteomic screen was done in cells transfected with tagged versions of TRIM5α or CypA and infected with VACV lacking the gene *C6L* (vΔC6; so that TRIM5α would not be degraded). The screen with CypA was done with or without CsA. Of the several viral proteins co-purified by either TRIM5α or CypA, L3 was the only structural protein enriched by both cellular proteins and the interaction with CypA was lost in the presence of CsA (Fig. 4a–c). To validate these observations, a codon-optimized VACV-WR *L3L* gene

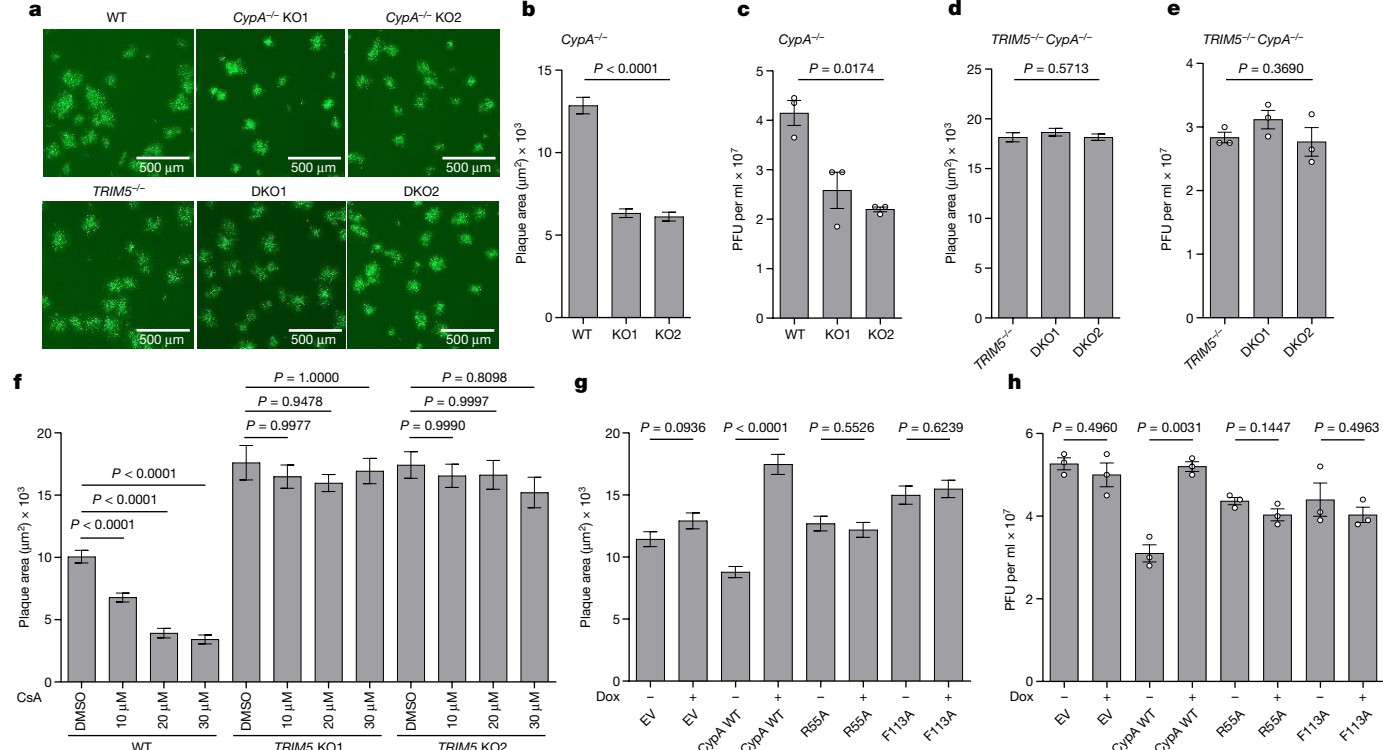

**Fig. 3 | CypA is a proviral factor in the presence of TRIM5. a**, Plaque image 24 hpi of T-REx-293 (WT) and derivative *CypA*[−/−] (KO1 and KO2) cells (top), and T-REx-293 *TRIM5*[−/−] and derivative *TRIM5*[−/−]*CypA*[−/−] (double knockout 1 (DKO1) or DKO2) cells (bottom) with VACV A5–GFP. **b**, Plaque area quantification of cells described in the top panels of **a**. $n \geq 107$ per condition. **c**, Infectious VACV titres following infection of T-REx-293 (WT) and derivative *CypA*[−/−] (KO1 and KO2) cells at 5 PFU per cell for 16 h. $n = 3$ per condition. **d**, Plaque area quantification of cells described in the bottom panels of **a**. $n \geq 236$ per condition. **e**, Infectious VACV titres following infection of T-REx-293 (WT) and derivative *TRIM5*[−/−], and *TRIM5*[−/−]*CypA*[−/−] (DKO1 and DKO2) cells at 5 PFU per cell for 16 h. $n = 3$ per condition. **f**, Plaque area quantification 2 days post-infection (dpi) of HeLa and derivative *TRIM5*[−/−] cells with VACV A5–GFP with the indicated concentrations of CsA. $n \geq 13$ per condition. **g**, Plaque area quantification 24 hpi of T-REx-293 *CypA*[−/−] cells engineered to inducibly express (+Dox) TAP-tagged WT CypA and mutants R55A and F113A with VACV A5–GFP. $n \geq 140$ per condition. **h**, Infectious VACV titres following infection of cells described in **g** at 5 PFU per cell for 16 h. $n = 3$ per condition. In **g**,**h**, cells were treated with 150 ng ml⁻¹ doxycycline from seeding for 24 h to express tagged proteins before harvest or infection. Data shown in **a**,**b**,**d**,**f**,**g** are representative of three independent experiments, and in **c**,**e**,**h** are from two independent experiments. Data from **b**–**f** were analysed using one-way Welch's ANOVA test ($P < 0.0001$ in **f**), and pairwise comparisons in **f** were performed using post-hoc Dunnett's T3 multiple comparisons test. Data from **g**,**h** were analysed using two-tailed unpaired Student's *t*-test. Analyses were performed on GraphPad Prism. Data are mean ± s.e.m. (**b**–**h**).

was expressed alongside either TRIM5α or CypA with or without CsA. Of note, L3 co-precipitated with both TRIM5α and CypA, and the latter interaction was prevented by CsA (Fig. 4d). The interactions between L3 and both CypA and TRIM5α were confirmed using endogenous L3 protein expressed during VACV infection and detected with an anti-L3 antibody[35]. Of note, catalytically defective CypA(R55A) and CypA(F113A) still co-precipitated L3 (Extended Data Fig. 4a), but were not proviral, indicating that the proviral activity of CypA is unlikely to merely reflect competition with TRIM5α for L3 binding.

To map where L3 binds to TRIM5α, the TRIM5α mutants shown in Fig. 1h were expressed alongside haemagglutinin (HA)-tagged L3, C6 or N1 (ref. 20). Immunoprecipitation showed that whereas C6 bound the TRIM5α RING domain, L3 required the SPRY domain, although this alone was insufficient for interaction (Fig. 4e). Possibly, the interaction requires TRIM5α dimerization to position SPRY domains appropriately, as is needed for retrovirus capsid binding[36–38]. To determine whether interactions were direct, the tagged proteins were expressed by in vitro transcription and translation followed by affinity purification. This showed that the TRIM5α–L3 interaction was likely to be direct, was greatly reduced in the absence of the SPRY domain and that L3 can dimerize (Extended Data Fig. 4b). TRIM5α enhances L3 dimerization in an E3 ubiquitin ligase activity-dependent manner (Extended Data Fig. 5a–d) and this is reversed in the presence of WT CypA, but not enzymatically inactive CypA mutants (Extended Data Fig. 5a,b).

Moreover, L3 undergoes a post-translational modification reminiscent of ubiquitylation in the presence of TRIM5α WT, but not the L19R mutant lacking E3 ubiquitin ligase activity, and this is also reversed by CypA (Extended Data Fig. 5e).

## L3 directs TRIM5α to virus factories

Next, the subcellular localization of TRIM5α and L3 were examined during VACV infection. Expression of tagged TRIM5α in T-REx-293 cells showed a broad cytoplasmic distribution; however, 10 h after infection with VACV vΔC6, the expression of TRIM5α was reduced (due to inhibition of host protein synthesis and protein turnover) and its location was restricted to virus factories that co-stained with L3 and cytoplasmic DNA (DAPI) (Extended Data Fig. 6a). To examine whether TRIM5α recruitment to factories was dependent on L3, *TRIM5*[−/−]*CypA*[−/−] cells complemented with TRIM5α by transfection were infected with the VACV strain vL3Li in which L3 protein expression is repressed, but were inducible by isopropyl β-D-1-thiogalactopyranoside (IPTG)[35]. This virus was used previously to demonstrate that L3 is critical for VACV infectivity, and although repression of L3 expression produced morphologically normal virions that bind to and enter cells, these virions fail to establish a productive infection due to a defect in early transcription[35]. This virus, grown with IPTG, was used to infect cells with or without IPTG. In uninfected cells, TRIM5α was broadly cytoplasmic with some

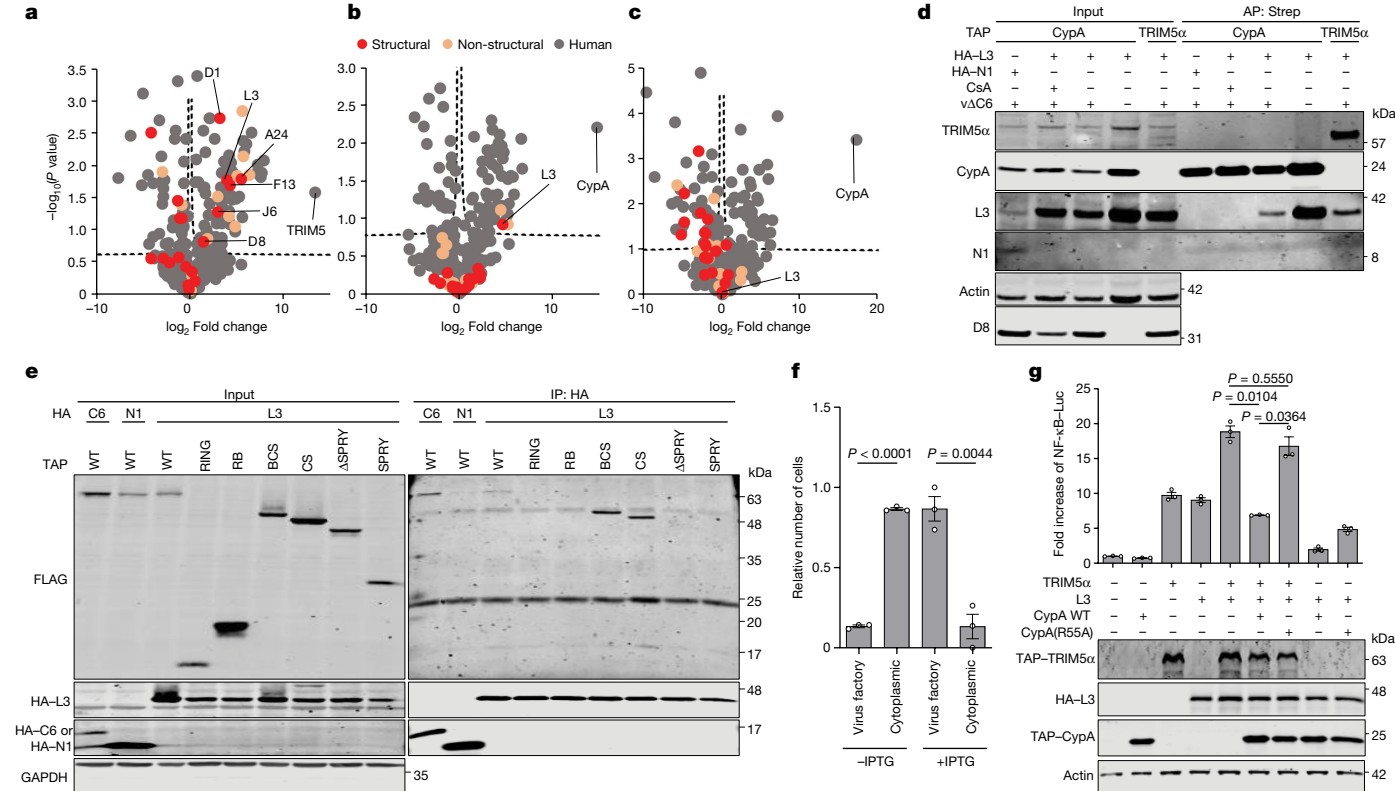

**Fig. 4 | Identification of TRIM5α and CypA interaction partners. a–c**, Volcano plots showing proteins co-purifying with TAP–TRIM5α (**a**), CypA (**b**) or CypA plus 20 μM CsA (**c**) and analysed by mass spectrometry. T-REx-293 *TRIM5⁻/⁻ CypA⁻/⁻* cells were transfected with TAP–TRIM5α or CypA and infected with vΔC6 at 3 PFU per cell for 10 h. Dashed lines indicate a false discovery rate of <0.05. The *P* values for L3 in **a**–**c** are 0.01763, 0.12212 and 0.93991, respectively. **d**, L3 precipitation with TRIM5α and CypA. T-REx-293 *TRIM5⁻/⁻ CypA⁻/⁻* cells were transfected with HA–L3 and TAP–TRIM5α or CypA ± CsA and infected with vΔC6 at 3 PFU per cell for 12 h, or mock-infected. **e**, Mapping of the L3 interaction domain. T-REx-293 *TRIM5⁻/⁻ CypA⁻/⁻* cells were co-transfected with HA–L3, HA–C6 or HA–N1 and TAP–TRIM5α WT and mutants (Fig. 1h). IP, Immunoprecipitation. **f**, TRIM5α localization for vL3Li ± IPTG. T-REx-293 *TRIM5⁻/⁻ CypA⁻/⁻* cells were transfected with TAP–TRIM5α for 12 h. Cells were infected with vL3Li at 2.5 PFU per cell for 10 h. The graph shows the relative number of cells in which TRIM5α is either localized in the viral factory only or also elsewhere in the cytoplasm ± IPTG. Data are from three independent experiments. *n* ≥ 36 per experiment.

**g**, TRIM5α and L3 stimulate NF-κB activation. T-REx-293 *TRIM5⁻/⁻ CypA⁻/⁻* cells were co-transfected with NF-κB–luciferase (Luc) and Renilla luciferase reporters alongside either empty vector, TRIM5α, L3 or CypA WT individually, or TRIM5α and L3 with CypA WT or R55A, or L3 with CypA WT or R55A. After 20 h, firefly luciferase activity was measured and normalized to Renilla luciferase. Fold induction is relative to empty vector. *n* = 3 per condition. TAP-tagged (**d**) or HA-tagged (**e**) proteins were precipitated using Strep-Tactin and anti-HA agarose beads and analysed alongside inputs by immunoblotting. Data shown are representative of three independent experiments, except **a**–**c**. In **a**–**c**, enriched proteins in the TRIM5 and CypA pulldowns were identified by comparison with empty vector using a two-sided Student's *t*-test and a permutation-based false discovery rate of <0.05. Data from **f**,**g** were analysed using one-way Welch's ANOVA test (where *P* < 0.0001 in **f** and *P* < 0.0001 in **g**), and pairwise comparisons were performed using post-hoc Dunnett's T3 multiple comparisons test. Data are mean ± s.e.m. (**f**,**g**).

puncta, whereas after infection in the presence of L3 (+IPTG), but not its absence (−IPTG), TRIM5α was recruited to virus factories (Fig. 4f and Extended Data Fig. 6b).

## L3 activates NF-κB signalling

In retroviruses, binding of TRIM5α to the virus capsid can trigger both premature uncoating of the capsid and activation of innate immune signalling[18,33,39]. To address whether TRIM5α interaction with the poxvirus protein L3 activated innate immunity, we utilized NF-κB reporter gene assays in cell lines lacking endogenous TRIM5α and CypA. Ectopic expression of TRIM5α induced NF-κB activation as noted by others[18,39,40], but L3 expression also activated this innate immune response, indicating that L3 can be recognized by innate immune sensors independent of TRIM5α (Fig. 4g). This was corroborated by IκBα (an inhibitor of NF-κB) degradation and upregulation of NF-κB-responsive genes upon the ectopic expression of L3 in *TRIM5⁻/⁻ CypA⁻/⁻* cells (Extended Data Fig. 6c,d). Furthermore, phosphorylation of TAK1 (p-TAK1) was investigated to explore the mechanism underlying TRIM5-independent L3-mediated NF-κB activation. Despite NF-κB activation, under the

conditions tested, p-TAK1 levels were not above background in control cells, suggesting that the activation is independent of p-TAK1 (Extended Data Fig. 6c). L3 and TRIM5α acted synergistically and NF-κB activation was enhanced in the presence of both proteins. The L3-mediated enhancement was reversed in the presence of CypA, but not by the catalytically inactive mutant R55A[41]. Of note, in the absence of TRIM5α, CypA could still antagonize L3-induced NF-κB activation (Fig. 4g). Next, the functional domains of TRIM5α that are needed for NF-κB activation, in the presence or absence of L3, were examined (Extended Data Fig. 6e). TRIM5α(L19R) and TRIM5α(ΔSPRY) were deficient in NF-κB activation in the absence of L3 and, unlike TRIM5α WT, did not augment L3-mediated pathway activation. By contrast, TRIM5α(N70A) was a more potent activator either alone or together with L3 (Extended Data Fig. 6e); this may be relevant to the ability of this mutant to restrict virus plaque size after low MOI, but not virus replication after high MOI (Fig. 2e,f).

## C6 and L3 are conserved in orthopoxviruses

C6 is a well-characterized multifunctional antagonist of innate immunity that co-precipitates with the IKKε and TBK1 adaptor proteins

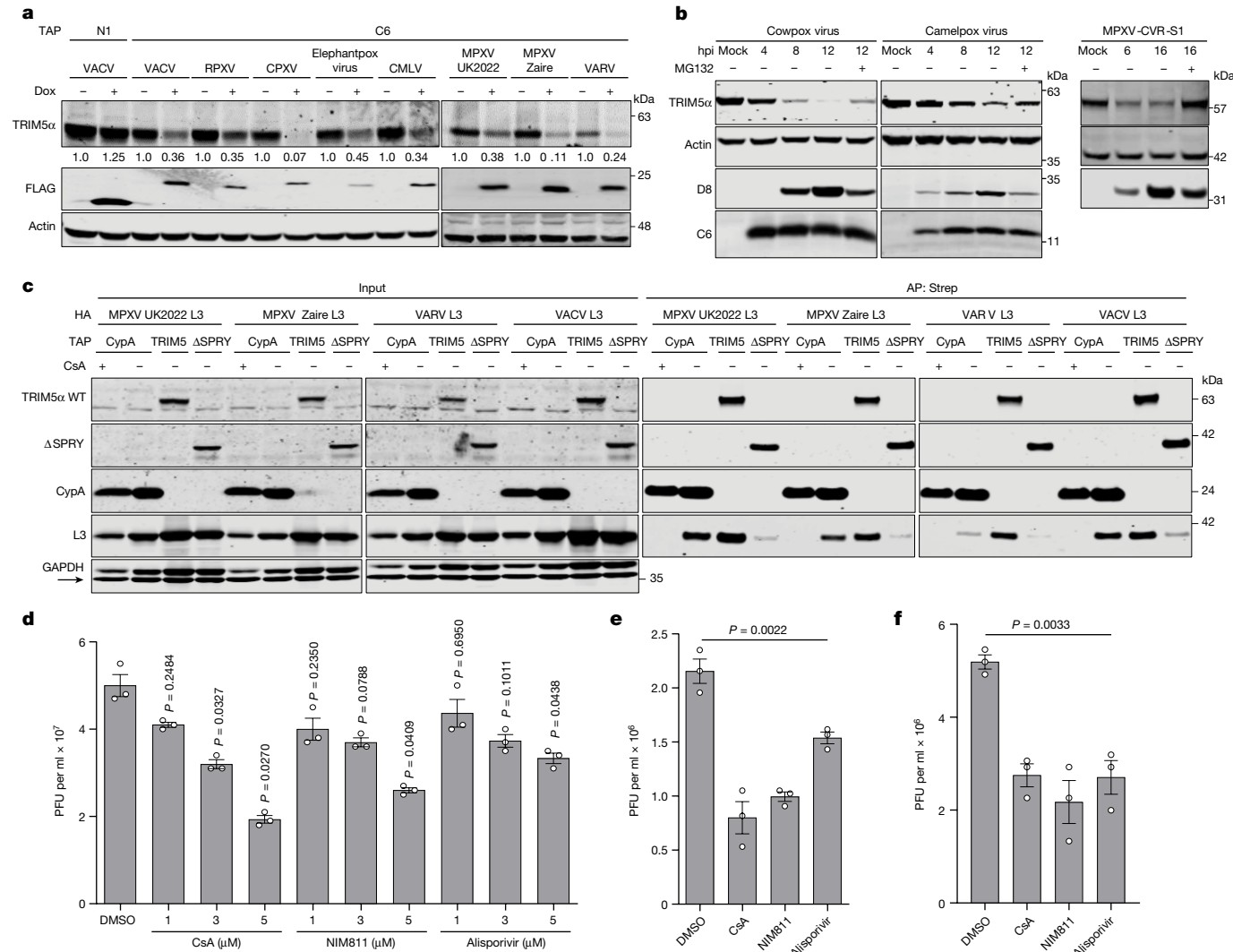

**Fig. 5 | Conservation of C6 and L3 in orthopoxviruses and the effect of CsA and derivatives. a**, Immunoblot showing TRIM5α in T-REx-293 cells inducibly expressing TAP–C6 proteins from VACV-WR, RPXV, CPXV, elephantpox virus, CMLV, MPXV UK2022, MPXV Zaire and VARV India 1967. The numbers indicate the band intensity of TRIM5α normalized to α-actin and relative to non-induced control. **b**, Immunoblot showing TRIM5α in HEK293T cells infected with CPXV or CMLV, and HeLa cells infected with MPXV-CVR-S1. Cells were infected at 5 PFU per cell. MG132 was added at 2 hpi. **c**, MPXV UK2022, MPXV Zaire, VARV and VACV L3 co-precipitation with TRIM5α and CypA. T-REx-293 *TRIM5⁻/⁻ CypA⁻/⁻* cells were transfected with HA–L3 and TAP–TRIM5α, ΔSPRY or CypA ± CsA 24 h before harvest. **d**, Effect of CsA and derivatives on infectious VACV titre following infection in T-REx-293 cells at 5 PFU per cell for 16 h. *n* = 3 per condition. **e**, Effect

of CsA and derivatives on infectious MPXV-CVR-S1 titres following infection in HFFF-TERT cells at 0.01 PFU per cell for 72 h. *n* = 3 per condition. **f**, Effect of CsA and derivatives on infectious MPXV-CVR-S1 titres following infection in HFFF-TERT cells at 5 PFU per cell for 18 h. *n* = 3 per condition. In **a**, cells were treated with 150 ng ml⁻¹ doxycycline for 16 h before harvest. Protein extracts (**b**) and inputs and AP proteins (**c**) were analysed by immunoblotting. In **d**–**f**, DMSO, CsA, NIM811 or alisporivir were added at 1 hpi. Data from **c** are representative of three independent experiments, whereas **a**,**b**,**d**–**f** are from two independent experiments. Data from **d**–**f** were analysed using one-way Welch's ANOVA test (where *P* < 0.0001 for **d**), and pairwise comparisons in **d** were performed using post-hoc Dunnett's T3 multiple comparisons test. Analyses were performed on GraphPad Prism. Data are mean ± s.e.m. (**d**–**f**).

TANK, SINTBAD and NAP1 to antagonize IRF3 activation[16], antagonizes type I IFN-induced signalling[15] and induces degradation of HDAC4 to inhibit type I IFN-mediated JAK–STAT signalling and IFN-stimulated gene expression[11]. Loss of C6 has been shown to reduce VACV virulence and enhance immunogenicity[42,43]. Highly conserved versions of C6 are encoded by most orthopoxviruses, including rabbitpox virus (RPXV; a VACV strain), CPXV, CMLV, elephantpox virus (a CPXV strain), MPXV and VARV (Extended Data Fig. 7a). The C6 orthologues from these viruses were expressed inducibly in cells and found to co-precipitate human TRIM5α (Extended Data Fig. 7b) and induce its degradation (Fig. 5a). CPXV C6 bound human TRIM5α and induced its degradation particularly well. The degradation of TRIM5α in a time-dependent manner was also observed after infection with RPXV,

CPXV, elephantpox virus, CMLV and MPXV-CVR-S1, a strain isolated during the 2022 global mpox epidemic (Fig. 5b and Extended Data Fig. 7c). The ability of all of these C6 orthologues to bind and degrade human TRIM5α is notable given that, except for VARV, the parent viruses are not endemic in humans and have natural reservoirs in other mammals.

Like C6, the L3 protein is highly conserved in orthopoxviruses, and orthologues from MPXV and VARV share more than 95% amino acid identity with VACV-WR L3 (Extended Data Fig. 8a). To test whether the L3 from MPXV and VARV could also bind to CypA and TRIM5α, L3 from MPXV (clade I), MPXV (clade IIb, representing the 2022 global mpox epidemic), VARV and VACV-WR were expressed in cells alongside CypA, TRIM5α or TRIM5α(ΔSPRY). Each L3 protein was able to co-precipitate

with CypA and TRIM5α, but not TRIM5α(ΔSPRY) (Fig. 5c). Moreover, in each case, the interaction with CypA was prevented by CsA.

CsA is a potent immunosuppressive cyclic decapeptide from which many non-immunosuppressive derivatives have been made, some of which have been tested in human clinical trials for other viral infections[44,45]. Therefore, we tested whether two of these compounds, alisporivir and NIM811, could also prevent the binding of CypA with the L3 protein from MPXV. Both compounds diminished the interaction of CypA with MPXV L3 (Extended Data Fig. 8b), just as for endogenous VACV L3 (Extended Data Fig. 8c). Finally, to test whether these compounds might restrict orthopoxvirus replication or spread, they were added to VACV-infected HEK293T cells (Fig. 5d). All of the drugs reduced virus yield significantly in a dose-dependent manner (Fig. 5d). However, CsA did not alter the virus yield in *CypA*[−/−] or *TRIM5*[−/−] cells (Extended Data Fig. 8d,e), confirming that its antiviral activity is dependent on CypA and TRIM5α, and that the antiviral activity was not due to cellular toxicity. These observations also extended to infection with MPXV-CVR-S1, in which the addition of CsA and non-immunosuppressive derivatives significantly reduced plaque size and infectious virus titre (Fig. 5e,f and Extended Data Fig. 8f). CsA also reduced the size of plaques formed by RPXV and CMLV (Extended Data Fig. 8g), in which the latter is closely related to VARV.

## Discussion

The study identifies TRIM5α as a restriction factor for orthopoxviruses. It shows that, as for retroviruses, the antiviral activity of TRIM5α is countered by the proviral activity of CypA, which in turn is antagonized by CsA and derivatives that prevent the binding of CypA to its viral target, the poxvirus capsid protein L3. L3 can dimerize, binds directly to the TRIM5α SPRY domain and is highly conserved in orthopoxviruses. The E3 ligase activity of TRIM5α is needed for antipoxviral activity and the prolyl isomerase activity of CypA is needed to antagonize this. In contrast to retroviruses, where interaction of TRIM5α with the viral capsid leads to capsid degradation, the interaction of TRIM5α with the poxvirus L3 protein enhances its dimerization, a novel consequence of TRIM5α recognition of the viral capsid protein. Dimerization is accompanied by biochemical modification, which is very reminiscent of ubiquitylation, and is antagonized by the prolyl isomerase activity of CypA. L3 is highly conserved in orthopoxviruses and we show that L3 from VACV, MPXV and VARV binds to human CypA and human TRIM5α, and the former interaction is prevented by CsA and derivatives. Given that human TRIM5α has exquisite specificity for different retroviral capsids, its interaction with L3 from non-human orthopoxviruses is notable. Furthermore, RPXV, MPXV and CMLV are restricted by CsA when grown in BS-C-1 cells, indicating that the African green monkey CypA and TRIM5α are likely to function in a similar manner to their human counterparts. The restriction of VACV replication by TRIM5α requires the RING domain, oligomerization and the SPRY domain. In addition, the ability of the N70A mutant to reduce viral titres after low, but not high, MOI suggests a second mechanism of viral restriction. Consistent with this idea, the L3 protein together with TRIM5α can induce NF-κB activation. Moreover, in the absence of TRIM5, L3 can still activate NF-κB, suggesting recognition of L3 via an additional unknown cellular factor acting at or upstream of IκBα. L3 is expressed late during infection, after NF-κB activation is suppressed by many viral inhibitors that are expressed early during infection, but L3 might be detected during uncoating of incoming virions unless encapsidated CypA blocks this. Nonetheless, the ability of a highly conserved poxvirus capsid protein to induce NF-κB activation might explain, in part, why VACV has evolved at least 15 different intracellular inhibitors of NF-κB activation[46,47].

This report also describes a second mechanism by which VACV and other orthopoxviruses antagonize TRIM5α. The viral protein C6 binds directly to the TRIM5α RING domain and induces its proteasome-dependent degradation. This is one of the first reports of a virus-encoded protein that induces TRIM5 degradation. Like L3, C6 is highly conserved in orthopoxviruses, and C6 orthologues from VACV, CPXV, CMLV, MPXV and VARV all bind to human TRIM5α and induce its degradation, despite most of these proteins deriving from viruses that are not endemic in humans. C6 is a small BCL-2-like protein that has multiple functions, including the inhibition of IRF3 signalling[16], JAK–STAT signalling via degradation of HDAC4 (refs. 11,15), degradation of the antiviral restriction factor HDAC5 (ref. 10) and, as shown here, degradation of TRIM5α. The degradation of TRIM5α that is induced by C6 is not via autoubiquitylation, because a mutant TRIM5α defective in autoubiquitylation is still degraded by C6. Rather, it seems likely that C6 engages additional E3 ubiquitin ligases and TRIM5α and facilitates the K63 ubiquitylation of the latter.

The role of CypA in antagonizing the antiviral activity of TRIM5α provides a route to antiviral drug development for orthopoxviruses such as MPXV and VARV. CsA and the non-immunosuppressive derivatives alisporivir and NIM811, interrupt the interaction of CypA and L3, and thereby reverse the proviral activity of CypA and enhance TRIM5α-mediated restriction. These compounds are therefore antiviral in the presence of CypA and TRIM5α and can restrict the replication and spread of orthopoxviruses. The potency of these drugs against viruses expressing C6, which degrades TRIM5α, will probably be enhanced if used in combination with drugs that disable C6. Finally, although the drugs brincidofovir and tecovirimat are licensed against VARV, these drugs target the viral proteins E9 (a DNA polymerase) or F13 (which is needed for virus spread[48,49]), respectively, so evolution of drug resistance will occur, as seen in the case of a patient with progressive vaccinia[50] and 22 patients infected with MPXV reported by the US Centers for Disease Control and Prevention during the current outbreak[51]. For tecovirimat, which does not block replication, a functional immune system is also needed to remove virus-infected cells alongside drug treatment; without this, infection can be fatal[52]. By contrast, CsA, alisporivir and NIM811 hinder virus replication by targeting a cellular protein, making the emergence of drug resistance difficult. Furthermore, both non-immunosuppressive derivatives have proceeded to at least phase II clinical trials[44,45], providing assurance in their safety. Therefore, clinical testing of these drugs against MPXV is warranted.

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

# Methods

## Cell lines and cell culture

The following cell lines were used in this study: human fetal foreskin fibroblast cells immortalized with human telomerase (HFFF-TERTs; male)[53], HeLa (American Type Culture Collection (ATCC) CCL-2), HEK293T (ATCC CRL-11268), T-REx-293 (Life Technologies), BS-C-1 (ATCC CCL-26) and RK13 (ATCC CCL-37). All cell lines were cultured in DMEM (Gibco), supplemented with 10% FBS (PAN Biotech) and 50 µg ml⁻¹ penicillin–streptomycin (Gibco). T-REx-293-derived cells that were stably transfected with pcDNA4/TO plasmids were further supplemented with 10 µg ml⁻¹ blasticidin (Thermo Fisher) and 100 µg ml⁻¹ zeocin (Gibco). HEK293T cells that were transduced to overexpress Myc-tagged proteins[10] were supplemented with 1 µg ml⁻¹ puromycin (Invivogen). A complete list of cell lines is described in Supplementary Table 1.

## Virus stocks

The viruses used in this study were VACV-WR and derivative viruses– v6/2 (ref. 54), vΔC6 (ref. 16), vTAP-C6 (ref. 55), vTAP-N1 (ref. 55), vA5L-EGFP[22] and vL3Li[35]–RPXV strain Utrecht, CPXV strain Brighton Red, elephantpox virus, CMLV strain CMS[56] and a strain of MPXV clade IIb isolated from a patient in Glasgow (UK) in 2022 (MPXV-CVR-S1). Viruses were grown in RK13 cells except for MPXV, which was grown in HFFF-TERTs. For vL3Li, infections were carried out in the presence of 25 µM IPTG. Infected cells were scraped from the culture flasks, collected by centrifugation and resuspended in 1 ml DMEM supplemented with 2% FBS (DMEM–2% FBS). Virus stocks were freeze-thawed three times and sonicated to release intracellular virus particles, and virus infectivity was measured by plaque assay on BS-C-1 cells.

## Plasmids

Single guide RNAs designed with UCSC Genome Browser to target either the genomic DNA encoding TRIM5, CypA or a small G protein signalling modulator in *Oryza sativa* Japonica Group (RICE), used as non-targeting, negative control, were annealed and cloned into the CRISPR–Cas9 plasmid px459 (#62988, Addgene) for the generation of knockout cell lines. cDNA for *TRIM5α*, *TRIM5γ* and *TRIM5δ* isoforms were amplified from HEK293T cells stimulated with IFNα and cloned into pcDNA4/TO-based plasmids with either FLAG or TAP tags; the TAP tag consisted of two copies of the Strep-tag II epitope and one copy of the FLAG epitope[57]. cDNA for *PPIA* was cloned into pcDNA4/ TO with DNA encoding a TAP tag at the 5′ end. *TRIM5α* and *PPIA* genes encoding mutants bearing amino acid substitutions were generated by using the Q5 Site-Directed Mutagenesis Kit (as directed by the manufacturer (New England Biolabs)) and were cloned into the pcDNA4/ TO plasmid with DNA encoding TAP tags at the 5′ end. The TRIM5α domain deletion mutants, RING (amino acids 1–87), RB (amino acids 1–129), BSC (amino acids 90–493), CS (amino acids 130–493) and ΔSPRY (amino acids 1–280), were generated by PCR from the TRIM5α expression plasmids described above. VACV-WR *L3L* codon-optimized for expression in human cells was synthesized by GenScript Biotech and cloned into the pcDNA3 and pcDNA4/TO plasmids with DNA encoding HA and TAP tags at the 5′ end, respectively. *L3L* orthologues from MPXV UK2022 (OP413717.1), MPXV Zaire (NP_536509.1) and VARV strain India 1967 (APR62813.1) were generated by site-directed mutagenesis of codon-optimized VACV *L3L*. pcDNA4/TO-based plasmids expressing the VACV-WR proteins C6, B14 and N1 have been previously described[16,55,58]. Codon-optimized VACV-WR *C1L* was synthesized by GeneArt (Thermo Fisher) and cloned into the pcDNA4/TO-based plasmid. *C6L* from orthopoxviruses was either amplified from viral genomic DNA (for CPXV, RPXV, elephantpox virus and CMLV), codon-optimized and synthesized by GenScript Biotech (for MPXV UK22), or generated by site-directed mutagenesis (for MPXV Zaire and VARV from codon-optimized MPXV UK2022 and VACV *C6L*, respectively) and cloned into the pcDNA4/TO plasmid encoding a TAP tag at the 5′ end. The NF-κB firefly luciferase

reporter and TK-Renilla luciferase plasmids were gifts from A. Bowie (Trinity College Dublin, Republic of Ireland). For in vitro transcription and translation assays, *HDAC5–HA*, *HDAC1–HA*, *HA–TRIM5α*, *HA–L3L*, *TAP–TRIM5α*, *TAP–TRIM5α(ΔSPRY)*, *TAP–CypA (PPIA)*, *TAP–L3L* and *TAP–C6L* were amplified from pcDNA4/TO plasmids either previously described[10,11] or from the above-described expression vectors and cloned into the pF3A WG (BYDV) plasmid (Promega). For the construction of vectors expressing a protein with different tags (HA or TAP), restriction enzymes were used to digest the open reading frame without tag from one vector and subclone it into the other vector that encodes the other tag. Oligonucleotides and primers used for cloning and sequencing are listed in Supplementary Table 2. A complete list of plasmids is described in Supplementary Table 3.

## Construction of knockout, complementation and overexpression cell lines

To generate *TRIM5* and *CypA* CRISPR–Cas9 knockout cell lines, HeLa (for *TRIM5⁻/⁻*) and T-REx-293 (*TRIM5⁻/⁻* and *CypA⁻/⁻*) clonal cells were transfected with pX459-derived plasmids expressing single guide RNAs targeting their respective genes. Transfected cells were selected with 1 µg ml⁻¹ puromycin and clonal cell lines obtained by limiting dilution were screened by immunoblotting and TOPO TA cloning (Thermo Fisher) and DNA sequencing. Complementation of knocked-out genes and overexpression of TRIM5 isoforms in T-REx-293 cells, which expresses the Tet repressor (TetR), were done by stable transfection of linearized pcDNA4/TO-based plasmids. Transfected cells were selected in DMEM–10% FBS supplemented with 10 µg ml⁻¹ blasticidin (Thermo Fisher) and 100 µg ml⁻¹ zeocin (Gibco).

## Virus replication, spread and infection assays

For virus growth assays, cells were infected with 0.01 or 5 PFU per cell at least three times. The inoculum was removed at 1 hpi and cells were washed once with warm DMEM and overlaid with DMEM–2% FBS. Cells were harvested at 16 and 24 hpi for 5 and 0.01 PFU per cell infections, respectively, and were freeze-thawed three times and sonicated. For MPXV growth assays, infected cells were harvested at 48 hpi for 0.01 PFU per cell infection. Infectious viral titres were determined by plaque assay on BS-C-1 cells. To assess viral spread, T-REx-293 or HeLa cell monolayers were infected with vA5L–eGFP[22] at 50–300 PFU per well and were overlaid with MEM–2% FBS supplemented with 2 mM L-glutamine, and 2% carboxymethylcellulose. Plaques were photographed at 24 hpi and the plaque size was measured using ImageJ. For immunoblotting, co-precipitation and immunofluorescence assays, cells were infected with the appropriate virus in DMEM–2% FBS. For infection with vL3Li, where indicated, 25 µM IPTG was added at the time of infection to induce expression of *L3L*. Virus adsorption was carried out at 37 °C for 1 h, and then cells were overlaid with DMEM–2% FBS.

## Transfection

Transfections were carried out using either TransIT-LT1 reagent (Mirus Bio) for HeLa cells and T-REx-293 (for reporter gene assay and mass spectrometry) or polyethylenimine (Polysciences) for HEK293T or T-REx-293 cells. Cells were seeded in the appropriate tissue culture dishes to reach 50% confluence on the day of transfection. The required amount of plasmid DNA and TransIT-LT1 (Mirus Bio) or polyethylenimine (2 µl per 1 µg DNA) were mixed into Opti-MEM (Gibco) (50 µl per 1 µg DNA) and incubated at room temperature for 25 min. Culture medium was replenished with DMEM–2% FBS and transfection mixtures were added dropwise to cells.

## Immunoblotting

Cells were scraped, washed twice with PBS (Sigma-Aldrich) and lysed with cell lysis buffer (50 mM Tris-HCl (pH 8.0), 150 mM NaCl, 1 mM EDTA, 10% (v/v) glycerol, 1% (v/v) Triton X-100 and 0.05% (v/v) NP-40), supplemented with protease (cOmplete Mini, Roche) and phosphatase

(PhosSTOP, Roche) inhibitors for 40 min on ice. Cell lysates were clarified by centrifugation at 13,000*g* for 10 min at 4 °C. Protein concentrations were determined using Pierce BCA protein assay (Thermo Fisher). Laemmli buffer (5×) was added to the samples and boiled at 100 °C for 10 min. Equal concentrations of protein samples were loaded onto SDS–polyacrylamide gels or NuPAGE 4–12% Bis-Tris precast gels (Invitrogen), separated by electrophoresis, and transferred to a nitrocellulose membrane (GE Healthcare). Membranes were blocked at room temperature with 5% (v/v) skimmed milk in TBS containing 0.1% (v/v) Tween-20 (TBS/T) for 1 h before incubation with the appropriate primary antibodies at room temperature for 1 h or at 4 °C overnight. After three 5-min washes with TBS/T, membranes were incubated with fluorophore-conjugated secondary antibodies (LI-COR Biosciences) at room temperature for 1 h. Membranes were washed three times with TBS/T and left to dry before imaging. Band intensities indicated in immunoblots and graphs were quantified using the Image Studio software (LI-COR Biosciences) and normalized to protein levels of loading control (α-actin or GAPDH). Primary and secondary antibodies used are listed in Supplementary Table 4.

## Co-precipitation assays

T-REx-293 or HEK293T cells were seeded in 10-cm dishes for transfection with indicated epitope-tagged plasmids, or for transfection followed by infection with the specified virus. Cells were harvested 24 h after transfection or 12 hpi on ice and washed twice with ice-cold PBS. For co-precipitation with Strep-Tactin Superflow agarose resin (IBA), 0.5% NP-40 in PBS was used as lysis and wash buffers. For immunoprecipitation with anti-HA agarose (Sigma-Aldrich), HA lysis/wash buffer (50 mM Tris-HCl (pH 6.8), 150 mM NaCl and 1% NP-40) was used. Lysis buffers were supplemented with protease (cOmplete Mini, Roche) and phosphatase (PhosSTOP, Roche) inhibitors. Lysis was carried out at 4 °C for 3 h. The insoluble fraction was collected by centrifugation at 13,000*g* for 15 min at 4 °C. Ten percent of the soluble fraction was collected as input and the remaining volume was incubated with the appropriate resin at 4 °C for 16 h. Protein-bound resins were washed three times with ice-cold wash buffer and proteins were eluted by boiling in 2× Laemmli buffer before analysis by SDS–PAGE and immunoblotting.

## In vitro transcription and translation

Equal amounts of pF3A-derived plasmids expressing proteins of interest were added to the TnT Sp6 high yield wheat germ protein expression system (Promega) according to the manufacturer's instructions and 10 μM CsA was added to the mixture where appropriate.

## Immunofluorescence

T-REx-293-derived cell lines were seeded on poly-D-lysine (Sigma-Aldrich)-coated sterile glass coverslips in six-well plates. Cells were either transfected or induced with 150 ng ml⁻¹ doxycycline (Melford) to express the protein of interest at least 12 h before infection at 2.5 PFU per cell with the appropriate VACV strain for 10 h. To harvest, cells were washed twice with warm PBS and fixed with 4% (v/v) paraformaldehyde for 15 min. Samples were quenched with 150 mM ammonium chloride for 5 min, washed twice with PBS and permeabilized with 0.1% Triton X-100 in PBS for 5 min. Cells were blocked with 10% (v/v) FBS in PBS for 30 min followed by staining with primary antibodies in 10% (v/v) FBS in PBS for 1 h at room temperature. Coverslips were washed three times with 10% (v/v) FBS in PBS for 5 min each and incubated with the appropriate AlexaFluor fluorophore-conjugated secondary antibodies (Molecular Probes) diluted in 10% (v/v) FBS in PBS supplemented with 2.5% of the corresponding normal serum (donkey or goat; Sigma-Aldrich) for 1 h in the dark at room temperature. Coverslips were washed three times with 10% (v/v) FBS in PBS and once with PBS before mounting onto glass slides with Mowiol 4-88 (Calbiochem) containing 0.5 μg ml⁻¹ 4′,6-diamidino-2-phenylindole (DAPI; Biotium). Images were acquired on a LSM700 confocal microscope (Zeiss) using the ZEN

system software (Zeiss). The antibodies used in immunofluorescence are listed in Supplementary Table 4.

## Reporter gene assay

T-REx-293 *TRIM5⁻/⁻* or *TRIM5⁻/⁻CypA⁻/⁻* cells were seeded in 96-well plates and used when 50% confluent. Plasmids expressing tagged proteins were co-transfected with 100 ng NF-κB–luciferase and 10 ng Renilla luciferase reporter plasmids using TransIT-LT1 (Mirus Bio). To induce protein expression, 150 ng ml⁻¹ doxycycline was added at the time of transfection for 24 h. Cells were then harvested in passive lysis buffer (Promega). Firefly luciferase activity was measured and normalized to Renilla luciferase control, and the fold induction was calculated relative to empty vector. Protein expression levels were determined by immunoblotting.

## Tandem affinity purification and mass spectrometry

T-REx-293 *TRIM5⁻/⁻CypA⁻/⁻* cells, complemented with either empty vector or TAP-tagged CypA by stable transfection, were seeded in three 10-cm dishes per sample and used at 50% confluency the next day. Cells were either transfected or induced to express TAP-tagged TRIM5α or CypA, respectively, for 12 h and then infected with vΔC6 at 3 PFU per cell for 10 h, in duplicate. Where appropriate, 20 μM CsA was added at 1 hpi. To harvest cells, they were scraped and washed twice with ice-cold PBS before lysis in 1 ml IP lysis buffer (0.5% NP-40 in PBS, supplemented with protease (cOmplete Mini, Roche) and phosphatase (PhosSTOP, Roche) inhibitors) at 4 °C for 2 h. The insoluble fraction was removed by centrifugation at 13,000*g* for 15 min at 4 °C. Of the soluble fraction, 10% was collected as input and the rest was incubated with anti-FLAG M2 agarose beads (Sigma) at 4 °C for 2 h. Protein-bound resins were washed three times with ice-cold wash buffer. Resin-bound proteins were eluted with 250 μg ml⁻¹ FLAG-peptide solution (Sigma-Aldrich) at 4 °C for 1.5 h and the eluted fractions were transferred to fresh tubes and incubated with Strep-Tactin Superflow agarose resin (IBA) at 4 °C for 8 h. Protein-bound resins were washed three times with ice-cold wash buffer. Of the resin-bound proteins, 25% were eluted by boiling in 2× Laemmli buffer before analysis by SDS–PAGE and Coomassie staining to validate the pulldown efficiency.

The remaining volumes were reduced, alkylated and trypsin-digested using the S-Trap protocol (ProtiFi). The resulting tryptic peptides were lyophilized and redissolved in 3% acetonitrile and 0.1% trifluoroacetic acid. Mass spectrometry data were acquired using a Q Exactive Plus coupled to an Ultimate 3000 RSLC nano UHPLC equipped with a 100-μm ID × 2-cm Acclaim PepMap precolumn (Thermo Fisher) and a 50-μm ID × 50 cm, 2-μm particle Acclaim PepMap RSLC analytical column. Loading solvent was 0.1% formic acid with analytical solvent A (0.1% formic acid) and solvent B (80% acetonitrile plus 0.1% formic acid). Samples were loaded at 5 μl min⁻¹ for 5 min before beginning the analytical gradient. The analytical gradient was 10–40% B over 67 min, increasing to 95% B by 80 min, and followed by a 4-min wash at 95% B and equilibration at 3% B for 12 min. Columns were held at 40 °C. Data were acquired in a data-dependent acquisition mode with the following settings: (1) first stage of mass spectrometry (MS1): 400–1,500 Th, 17,500 resolution, $1 × 10^5$ automatic gain control target and 250-ms maximum injection time. (2) MS2: quadrupole isolation at an isolation width of $m/z$ 3.0 and higher-energy C-trap dissociation fragmentation (normalized collision energy 30). Dynamic exclusion was set for 30 s and MS2 fragmentation was triggered on precursors of $3.2 × 10^4$ counts and above. Raw files were processed using MaxQuant (v.2.0.1.0) and searched against a human UniProt database (downloaded on 23 September 2020) and a UniProt vaccinia virus database (downloaded on 10 March 2022). Carbamidomethyl (C) was set as a fixed modification with oxidation (M) and acetyl (protein N terminus) as variable peptide modifications. Additional analyses were conducted using Perseus software (version 1.6.2.1)[59], obtaining sample relative protein abundance after potential contaminants and reverse protein identifications were removed, and

data imputation to account for missing values in individual samples. Comparisons between empty vector and TRIM5 or CypA pulldowns were performed with a two-sided Student's $t$-test with significance cut-offs defined by a permutation-based false discovery rate of 0.05, an S0 parameter of 0.1 and 250 permutations (implemented in Perseus).

The mass spectrometry proteomics data have been deposited to the iProX repository[60], a ProteomeXchange Consortium (http://www.proteomexchange.org/) partner, with the dataset identifiers IPX0005650001 and PXD039094, respectively.

### RT–qPCR
Quantitative PCR with reverse transcription (RT–qPCR) analysis of NF-κB-responsive genes was undertaken as previously described[61]. In brief, the $TRIM5^{-/-}CypA^{-/-}$ cell line was modified to express inducibly either VACV TAP-tagged-L3 or empty vector. These cells were seeded in 12-well plates at a density of $6 \times 10^5$ cells per well; the next day, these cells were incubated in medium without serum for 3 h and then either mock-treated or treated with doxycycline (150 ng ml$^{-1}$) for 4 h in triplicate. Total RNA was extracted from the cells and cDNA was synthesized by reverse transcription using oligo-dT primers (Thermo Fisher). The mRNA levels of the NF-κB-responsive genes *NFKBIA*, *CCL2*, *CXCL8*, *CXCL10*, *IL6* and the housekeeping control gene *GAPDH* were measured by qPCR using Fast SYBR Green master mix (Thermo Fisher) and a ViiA 7 real-time PCR machine (Thermo Fisher). The fold induction of the mRNA levels was calculated by the $2^{-\Delta\Delta Ct}$ method using induced T-REx-293 empty vector and *GAPDH* as the internal control. The primers used in the qPCR are listed in Supplementary Table 2.

### Alignment
Identifiers for the *C6L* orthologues in orthopoxvirus genomes are as follows: VACV (YP_232904.1), RPXV (AAS49727.1), CPXV (NP_619819.1), CMLV (NP_570410.1), MPXV UK 2022 (UWM73173.1), MPXV Zaire (NP_536441.1) and VARV major strain India 1967 (P0DSX3.1). Identifiers for the *L3L* orthologues in orthopoxvirus genomes are as follows: VACV (YP_232972.1), RPXV (AAS49792.1), CPXV (ADZ24099.1), CMLV (NP_570478.1), MPXV UK 2022 (UWM73237.1), MPXV Zaire (NP_536509.1) and VARV major strain India 1967 (APR62813.1). Amino acid sequences were aligned using Clustal Omega.

### Statistical analysis
Data are presented as means ± s.e.m., and statistical significance was analysed in Prism (GraphPad) using Welch's ANOVA test and followed by post-hoc Dunnett's T3 multiple comparisons test where indicated or two-tailed unpaired $t$-test with Welch's correction. The exact $P$ values are shown in each figure and the horizontal bars indicate the samples being compared. The number of repeats and the values of $n$ in each experiment are indicated in the respective figure legends; $n$ represents the number of biological replicates.

### Reporting summary
Further information on research design is available in the Nature Portfolio Reporting Summary linked to this article.

### Data availability
All data from this study including supplementary material will be freely available. Proteomic data generated from label-free mass spectrometry are uploaded in the iProX repository, a ProteomeXchange Consortium partner, with the dataset identifiers IPX0005650001 and PXD039094, respectively. Orthopoxvirus nucleotide sequences cited in this study are available on NCBI GenBank: VACV (YP_232972.1 and YP_232904.1), RPXV (AAS49792.1 and AAS49727.1), CPXV (ADZ24099.1 and NP_619819.1), CMLV (NP_570478.1 and NP_570410.1), MPXV UK 2022 (UWM73237.1 and UWM73173.1), MPXV Zaire (NP_536509.1 and NP_536509.1) and VARV major strain India 1967 (APR62813.1 and P0DSX3.1). Source data are provided with this paper.

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

**Acknowledgements** We thank B. Moss (NIH) for the recombinant virus vL3Li and for the antibody to the L3 protein; C. Damaso (Federal University of Rio de Janeiro) and M. P. Weekes (CIMR) for helpful discussions; and R. Antrobus (CIMR Proteomics facility) for technical support with mass spectrometry. This work was supported in part by grants from the Wellcome Trust (no. 090315), the Isaac Newton Trust (no. 21.23(e)), University of Cambridge, BBSRC grant BB/X011542/1, MRC grant MR/W025590/1 and a PhD studentship to Y.Z. from the Department of Pathology, University of Cambridge.

**Author contributions** Y.Z., Y.L., J.D.A. and G.L.S. designed the experiments. Y.Z., Y.L., S.R. and M.S. performed the experiments. Y.Z., Y.L. and J.D.A. analysed the data. Y.Z., Y.L., J.D.A. and G.L.S. wrote and edited the manuscript. G.L.S. and J.D.A. supervised the project. G.L.S. acquired funding.

**Competing interests** The authors declare no competing interests.

**Additional information**
**Correspondence and requests for materials** should be addressed to Jonas D. Albarnaz or Geoffrey L. Smith.

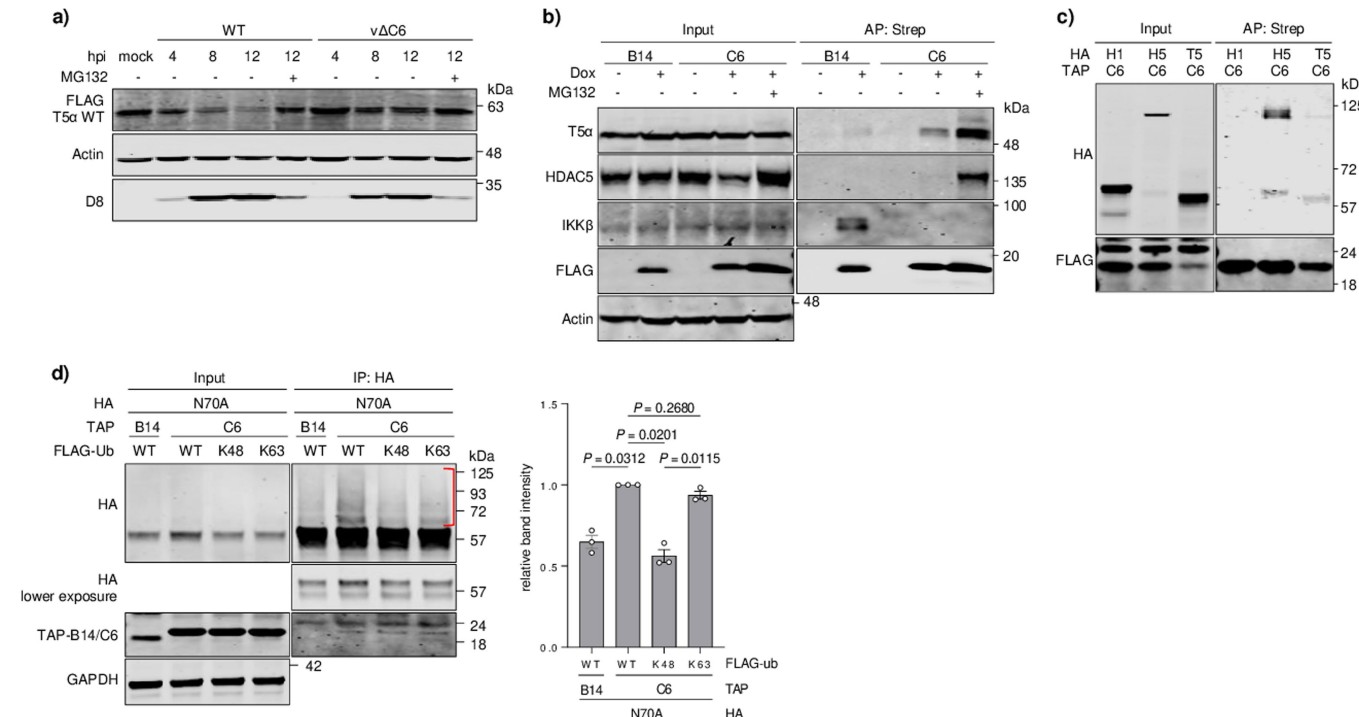

**Extended Data Fig. 1 | VACV protein C6 induces proteosomal degradation of TRIM5α. a)** Immunoblots showing TRIM5α abundance in T-REx-293 TRIM5[-/-] cells complemented with inducible expression (+dox) of TAP-tagged TRIM5α and infected with WT or vΔC6. **b)** C6 and TRIM5α co-precipitate. HEK293T cell lines were induced (+Dox) to express TAP-tagged C6 or B14 +/- MG132. **c)** C6 and TRIM5α interact. HA-tagged TRIM5, HDAC1 and HDAC5 were co-expressed with TAP-tagged C6 using the wheat germ transcription/translation system. HDAC1 and HDAC5 were used as negative and positive controls for direct interaction with C6, respectively. **d)** C6-mediated ubiquitylation of TRIM5α. T-REx-293 TRIM5[-/-]CypA[-/-] cells were transfected to co-express TAP-tagged B14 or C6, FLAG-tagged WT/K48/K63 ubiquitin and HA-tagged TRIM5α N70A.

Red bracket indicates modified TRIM5α N70A. Graph shows band intensity of higher molecular mass TRIM5α N70A indicated by the red bracket from three independent experiments, normalised to the unmodified TRIM5α N70A band shown by the HA lower exposure blot and relative to the sample transfected with TAP-tagged C6, FLAG-tagged WT ubiquitin and HA-tagged TRIM5α N70A. n = 3/condition. Data was analysed using One-Way Welch's ANOVA test (p = 0.0031) and pairwise comparisons was performed using Dunnett's T3 multiple comparisons test. Mean ± s.e.m. Protein extracts in **a)** and inputs and AP proteins in **b-d)** were analysed by SDS-PAGE and immunoblotted for the indicated epitope/protein. Data shown in **a)**, **b)** and **d)** are representative of three independent experiments and **c)** is from two.

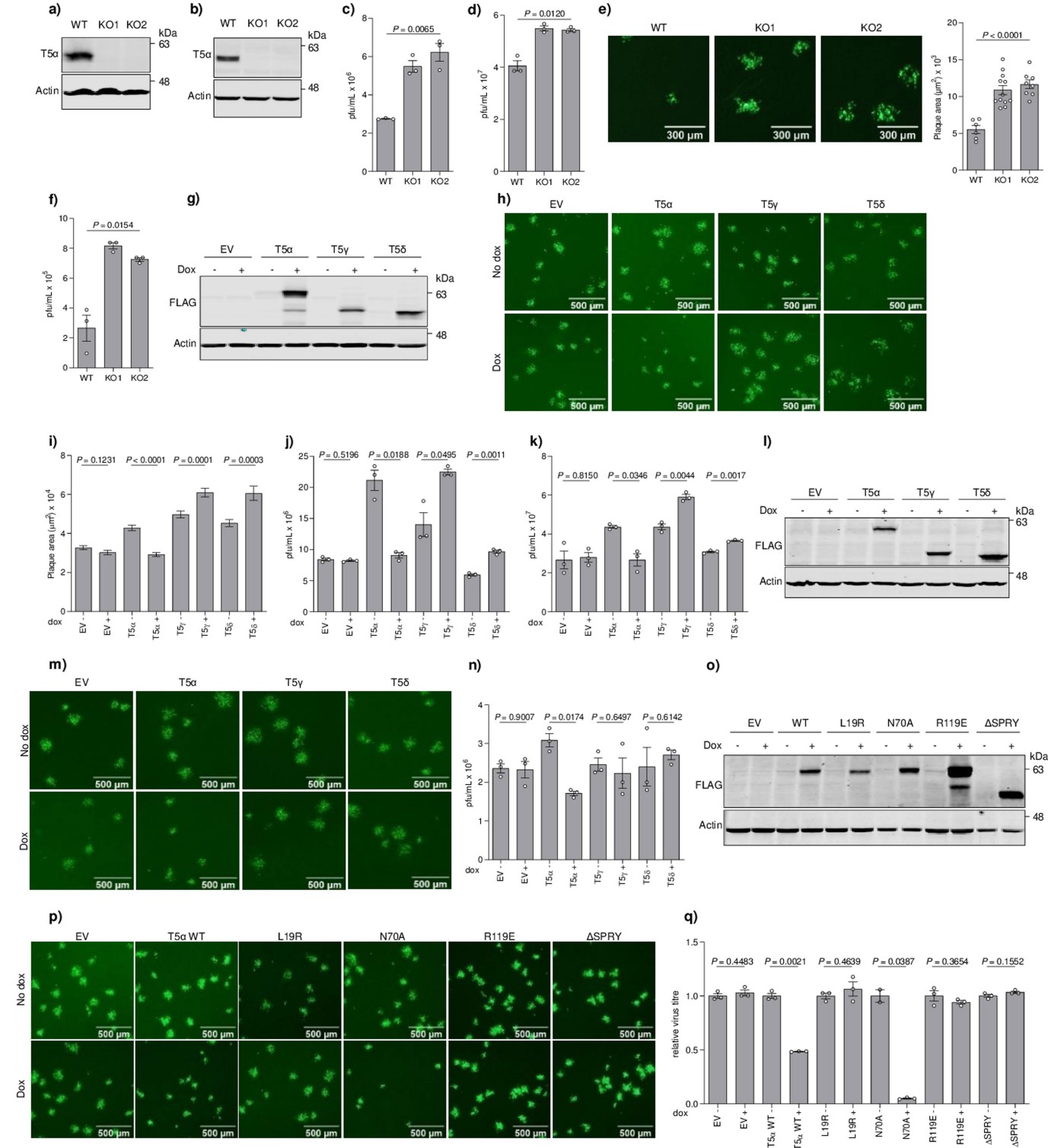

**Extended Data Fig. 2** | See next page for caption.

**Extended Data Fig. 2 | TRIM5α is a VACV restriction factor. a)** Immunoblot showing TRIM5 levels in T-REx-293 (WT) and derivative TRIM5$^{-/-}$ (KO1/2) cells. **b)** Immunoblot showing TRIM5 levels in HeLa (WT) and derivative TRIM5$^{-/-}$ (KO1/2) cells. **c)** Infectious VACV titres following infection of cells described in **b)** at 5 PFU/cell for 16 h. n = 3/condition. **d)** Infectious VACV titres following infection of cells described in **a)** at 0.01 PFU/cell for 24 h. n = 3/condition. **e)** Plaque image (left) and plaque area quantification (right) 24 hpi of cells indicated in **c)** with VACV A5-GFP. n ≥ 6/condition. **f)** Infectious VACV titres following infection of cells described in **b)** at 0.01 PFU/cell for 24 h. n = 3/condition. **g)** Immunoblot showing inducible expression (+Dox) of FLAG-tagged TRIM5α/γ/δ isoforms, in T-REx-293 cells. **h)** Plaque image 24 hpi of cells indicated in **g)** with VACV A5-GFP. **i)** Plaque area quantification of **h)**. n ≥ 48/condition. **j)** Infectious VACV titres following infection of cells described in **g)** at 0.01 PFU/cell for 24 h. n = 3/condition. **k)** Infectious VACV titres following infection of cells described in **g)** at 5 PFU/cell for 16 h. n = 3/condition. **l)** Immunoblot showing inducible expression (+Dox) of TAP-tagged TRIM5α/γ/δ isoforms, introduced into T-REx-293 TRIM5$^{-/-}$ cells. **m)** Plaque image 24 hpi of cells indicated in **l)** with VACV A5-GFP. **n)** Infectious VACV titres following infection of cells described in **l)** at 0.01 PFU/cell for 24 h. n = 3/condition. **o)** Immunoblot showing inducible expression (+Dox) of TAP-tagged TRIM5α mutants, L19R, N70A, R119E and ΔSPRY introduced into T-REx-293 TRIM5$^{-/-}$ cells. **p)** Plaque images of VACV A5-GFP infection of cells described in **o)** for 24 h. **q)** Infectious VACV titres following infection of cells described in **o)** at 0.01 PFU/cell for 24 h. Viral titres were normalised to the respective uninduced (-dox) controls. n = 3/condition. In **g-n)** cells were treated with 150 ng/mL doxycycline from seeding for 24 h to express tagged proteins before harvest or infection. In **o-q)** doxycycline was added at 24 h after seeding. In **a)**, **b)**, **g)**, **l)** and **o)** protein extracts were analysed by SDS-PAGE and immunoblotted for the indicated epitope/protein. Data shown in **d)**, **h)**, **i)**, **m)** and **p)** are representative of three independent experiments, and in **b)**, **e)**, **f)**, **j)**, **k)** and **q)** are from two. Data from **b)** and **d-f)** were analysed using One-Way Welch's ANOVA test. Data from **i-k)**, **n)** and **q)** were analysed using two-tailed unpaired Student's t-test. Analyses were performed on GraphPad Prism. Mean ± s.e.m.

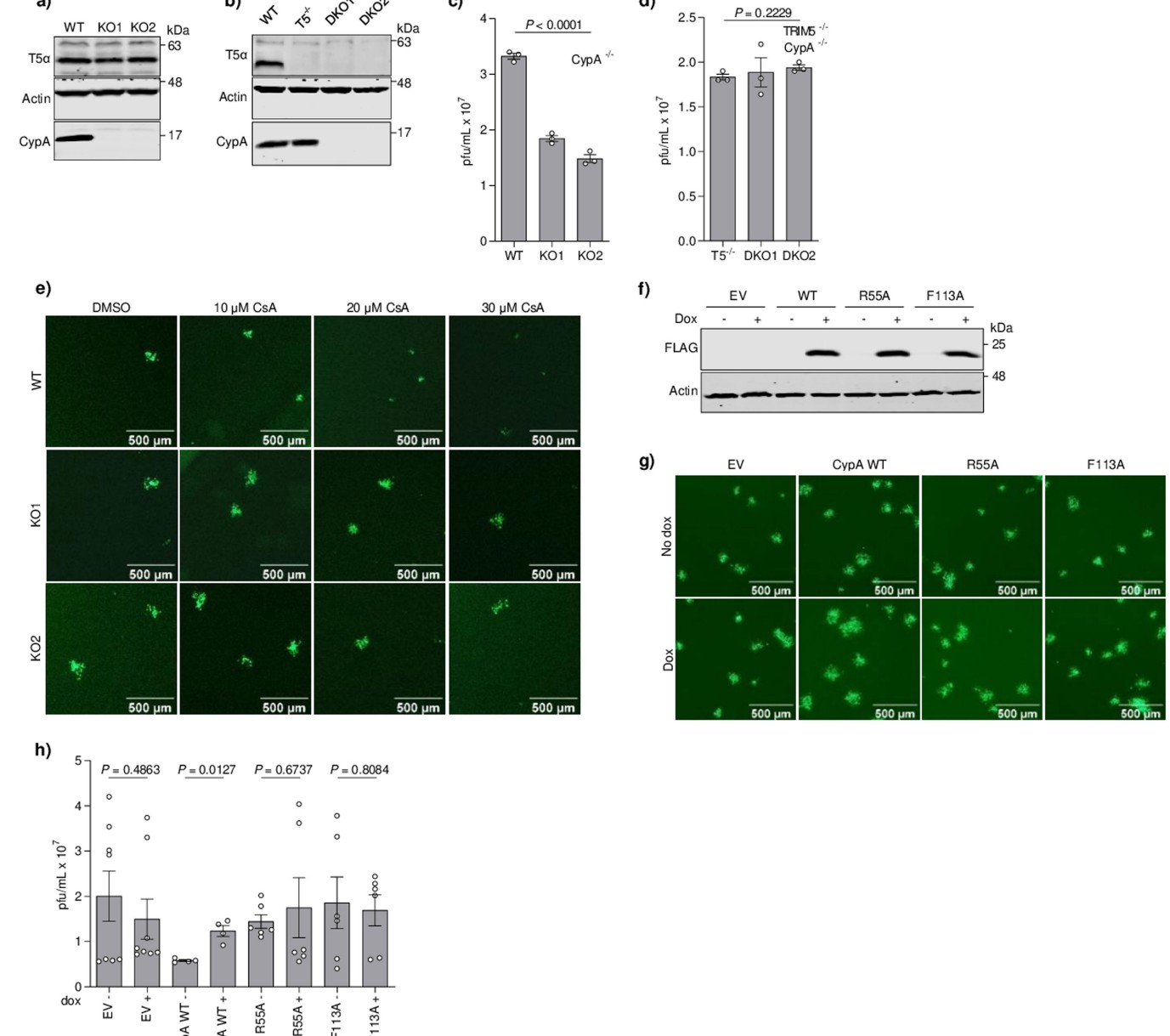

**Extended Data Fig. 3 | CypA is a pro-viral factor in the presence of TRIM5.**
**a)** Immunoblot showing CypA levels in T-REx-293 (WT) and derivative CypA[-/-] (KO1/2) cells. **b)** Immunoblot showing TRIM5 and CypA levels in T-REx-293 (WT) and derivative TRIM5[-/-], and TRIM5[-/-]CypA[-/-] (DKO1/2) cells. **c)** Infectious VACV titres following infection of cells described in **a)** at 0.01 PFU/cell for 24 h. n = 3/ condition. **d)** Infectious VACV titres following infection of cells described in **b)** at 0.01 PFU/cell for 24 h. n = 3/condition. **e)** Plaque image of Fig. 3f. Plaques were selected at random, imaged using AxioVision 4.8 and measured using ImageJ. **f)** Immunoblot showing inducible expression (+Dox) of TAP-tagged WT CypA and mutants R55A and F113A, introduced into T-REx-293 CypA[-/-] cells. **g)** Plaque image 24 hpi of cells indicated in **f)** with VACV A5-GFP. **h)** Infectious VACV titres following infection in cells described in **f)** at 0.01 PFU/cell for 24 h. n = 8/ condition for EV, n = 4/condition for CypA WT and n = 6/condition for R55A and F113A. Data shown in **e)** and **g)** are representative of three independent experiments, and **c)**, **d)** and **h)** are from two. Data from **c)** and **d)** were analysed using One-Way Welch's ANOVA test. Data from **h)** was analysed using two-tailed unpaired Student's t-test. Analyses were performed on GraphPad Prism. Mean ± s.e.m.

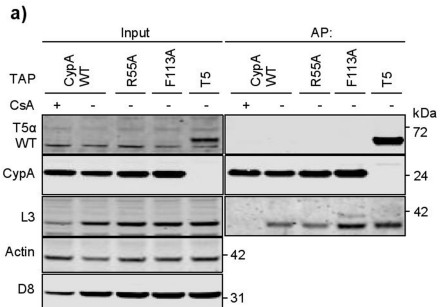 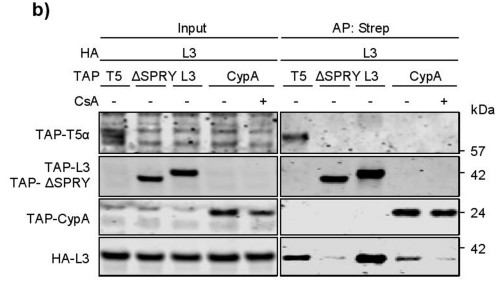

**Extended Data Fig. 4 | Identification of TRIM5α and CypA interaction partner(s). a)** L3 co-precipitation with CypA WT and mutants. T-REx-293 TRIM5$^{-/-}$CypA$^{-/-}$ cells were transfected with TAP-tagged TRIM5α, CypA WT +/− CsA or CypA mutants, R55A and F113A, and infected with vΔC6 at 3 PFU/cell for 12 h. **b)** L3 dimerisation and direct interaction with CypA and TRIM5α shown using cell-free transcription and translation system. TAP-tagged L3, TRIM5α WT, ΔSPRY and CypA were co-expressed with HA-tagged L3. In **a)** and **b)** TAP-tagged proteins were precipitated using Strep-Tactin. Inputs and AP proteins were analysed by SDS-PAGE and immunoblotted for the indicated epitope/protein. α-Actin and D8 were controls for loading and viral infection, respectively. Data from **a)** are representative of three independent experiments, and **b)** is from two.

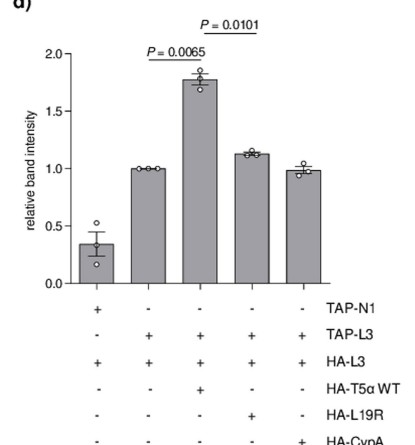

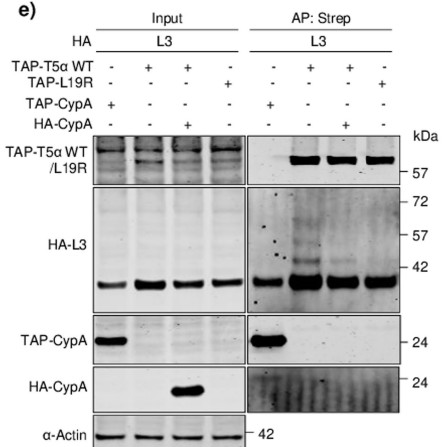

**Extended Data Fig. 5** | See next page for caption.

**Extended Data Fig. 5 | L3 dimerisation. a)** L3 dimerisation is stabilised by TRIM5α and reversed by CypA WT. T-REx-293 TRIM5$^{-/-}$CypA$^{-/-}$ cells were co-transfected with TAP-tagged L3 or N1, HA-tagged L3, TRIM5α WT, ZAP-L, CypA WT, R55A or F113A with or without 5 μM CsA for 24 h. **b)** Relative band intensity of HA-tagged L3 in **a)** AP samples. Graph shows band intensity of HA-tagged L3 from three independent experiments, normalised to TAP-tagged L3 and relative to the sample transfected with TAP-tagged and HA-tagged L3. n/condition is as indicated. Data points were excluded in samples where n < 3 due to high background noise causing inaccuracy in quantification. **c)** Stabilisation of L3 dimerisation by TRIM5α requires its E3 ubiquitin ligase activity. T-REx-293 TRIM5$^{-/-}$CypA$^{-/-}$ cells were co-transfected with TAP-tagged L3 or N1 and HA-tagged L3, TRIM5α WT, L19R or CypA WT. **d)** Relative band intensity of HA-tagged L3 in **c)** AP samples. Graph shows band intensity of HA-tagged L3 from three independent experiments, normalised to TAP-tagged L3 and relative to the sample transfected with TAP-tagged and HA-tagged L3. n = 3/condition. Data were analysed using were analysed using One-Way Welch's ANOVA test ($p$ = 0.0007) and pairwise comparisons were performed using Dunnett's T3 multiple comparisons test on GraphPad Prism. Mean ± s.e.m. **e)** Modification of L3 by TRIM5α WT requires its E3 ubiquitin ligase activity and is reversed by CypA WT. T-REx-293 TRIM5$^{-/-}$CypA$^{-/-}$ cells were co-transfected with TAP-tagged TRIM5α WT, L19R or CypA WT and HA-tagged L3 or CypA. Tagged viral proteins were affinity-purified using Strep-Tactin beads. Inputs and AP proteins in **a)**, **c)** and **e)** were analysed by SDS-PAGE and immunoblotted for the indicated epitope/protein. Data shown are representative of three independent experiments.

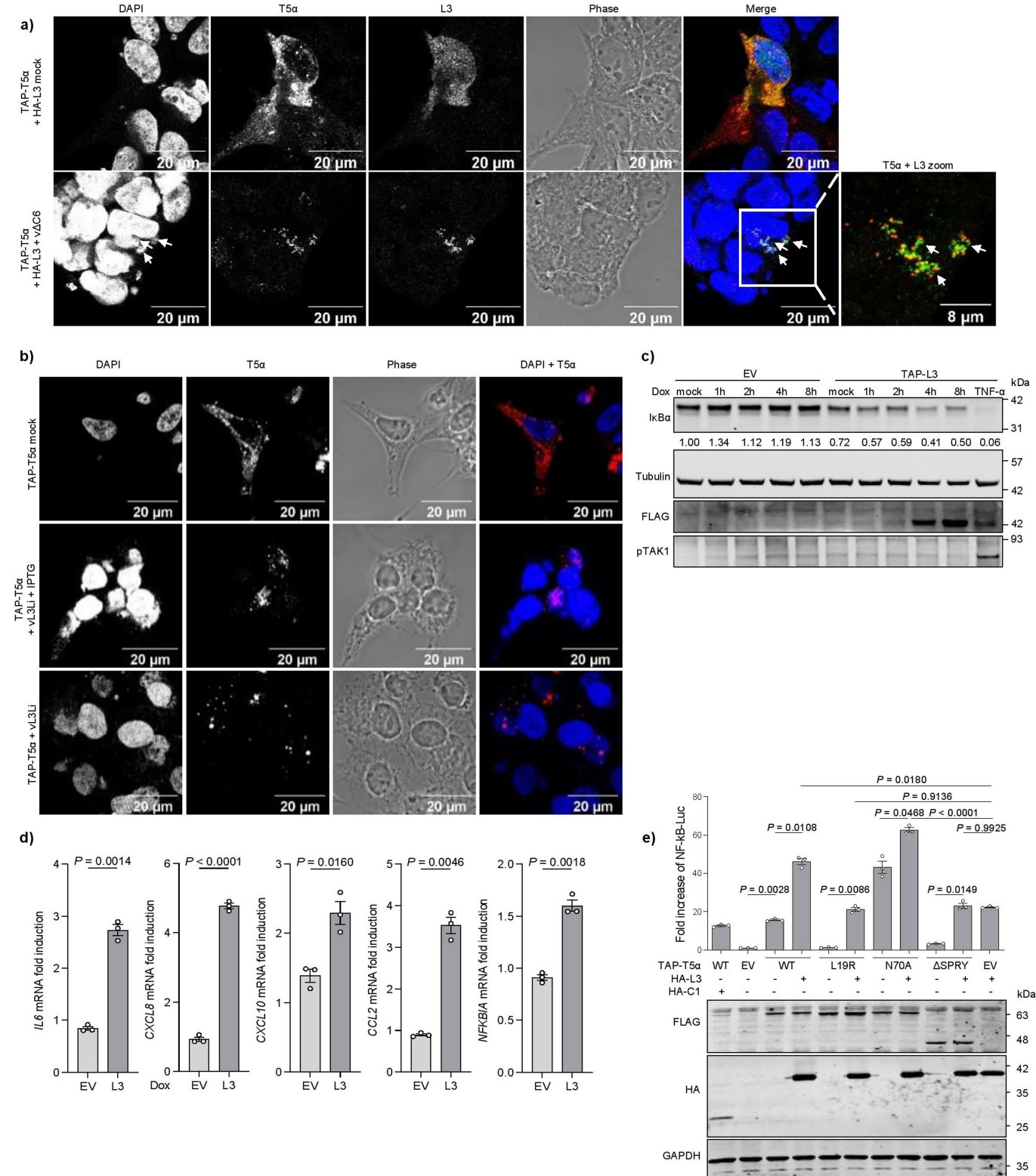

**Extended Data Fig. 6** | See next page for caption.

**Extended Data Fig. 6 | Localisation of L3 and activation of innate immunity.**
**a)** TAP-tagged TRIM5α and HA-tagged L3 localisation during vΔC6 infection.
T-REx-293 TRIM5$^{-/-}$CypA$^{-/-}$ cells were transfected with TAP-tagged TRIM5α and
HA-tagged L3 and infected with vΔC6 at 2.5 PFU/cell for 10 h. Cells were stained
with DAPI (blue) and antibodies against FLAG (red) and HA (green). White
arrows indicate viral factories. Images were acquired using a Zeiss LSM 700
confocal microscope. **b)** TRIM5α localisation during vL3Li +/- IPTG. T-REx-293
TRIM5$^{-/-}$CypA$^{-/-}$ cells were transfected with TAP-tagged TRIM5α for 12 h. Cells
were infected with vL3Li, whose L3 expression is induced only upon IPTG
treatment, at 2.5 PFU/cell for 10 h. Cells were stained with DAPI (blue), and
antibodies against FLAG (red). Images were acquired using Zeiss LSM 700.
Quantitative analysis shown in Fig. 4f. **c)** IκBα degradation following TAP-
tagged L3 expression. T-REx-293 TRIM5$^{-/-}$CypA$^{-/-}$ cells stably transfected with
TAP-tagged EV or L3 were serum starved for 4 h prior to doxycycline treatment
for the indicated times. TNF-α stimulation was for 30 min. **d)** Fold induction of
*IL6*, *CXCL8*, *CCL2*, *NFKBIA* and *CXCL10* mRNAs. T-REx-293 TRIM5$^{-/-}$CypA$^{-/-}$ cells
stably transfected with TAP-tagged EV or L3 were induced with doxycycline for
4 h before harvest and analysis by RT-qPCR. n = 3/condition. **e)** L3 and TRIM5α
WT or mutant N70A stimulate NF-κB synergistically. TRIM5$^{-/-}$CypA$^{-/-}$ cells were
co-transfected with 100 ng NF-κB-luciferase and 10 ng *Renilla* luciferase
alongside either EV, L3, TRIM5α WT or mutants L19R, N70A and ΔSPRY,
individually, or TRIM5α WT or mutants with L3 or C1. Cells were harvested 24 h
after transfection, firefly luciferase activity was measured and normalised to
*Renilla* luciferase. Fold induction is relative to EV. n = 3/condition. Data shown
in **a)**, **b)**, and **e)** are representative of three independent experiments and **c)** and
**d)** are from two. Data from **d)** was analysed using two-tailed unpaired Student's
t-test on GraphPad Prism. Data from **e)** was analysed using One-Way Welch's
ANOVA test ($p < 0.0001$) and pairwise comparisons were performed using
Dunnett's T3 multiple comparisons test. Mean ± s.e.m.

**a)**

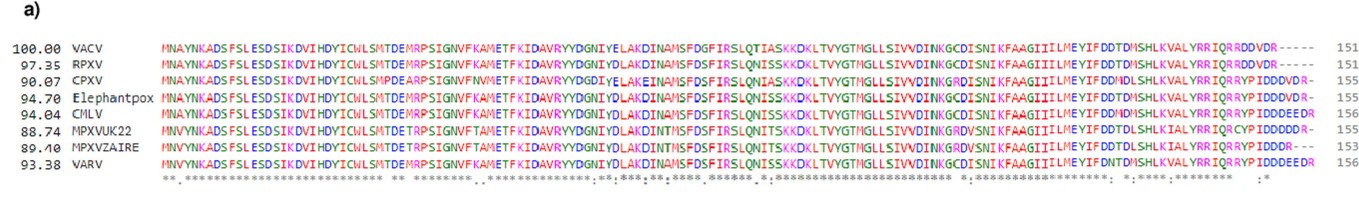

**b)**

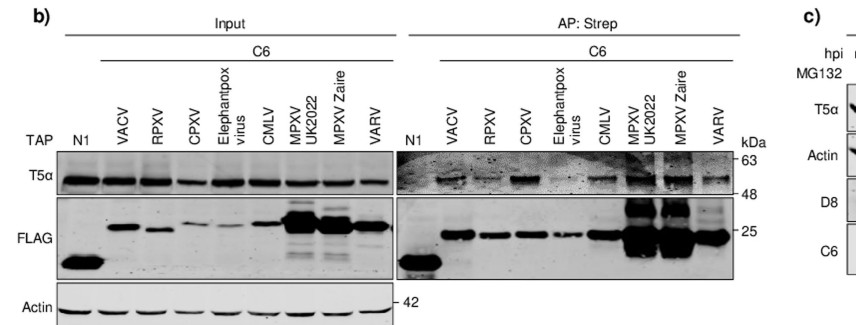

**c)**

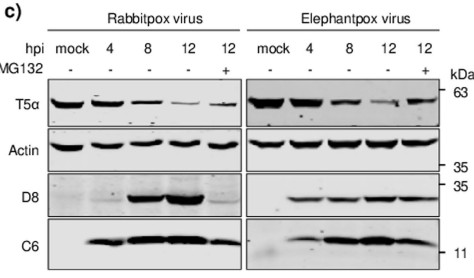

**Extended Data Fig. 7 | Conservation of C6 in orthopoxviruses. a)** Amino acid sequence alignment of orthopoxvirus C6 proteins created by Clustal Omega. Amino acids were colour-coded according to their properties. Asterisks (*) indicate fully conserved residues, colons (:) indicate amino acids with strongly similar properties and dots (.) indicate amino acids with weakly similar properties. Numbers on the left indicate % amino acid identity compared to VACV WR C6. **b)** Orthopoxvirus C6 co-precipitation with endogenous TRIM5α in T-REx-293 cells inducibly expressing TAP-tagged C6 proteins from VACV WR, rabbitpox virus (RPXV), cowpox virus strain Brighton Red (CPXV), elephantpox virus, camelpox virus (CMLV) strain CMS, monkeypox virus (MPXV) strain UK2022, MPXV Zaire and variola major virus (VARV) strain India 1967. TAP-tagged proteins were affinity-purified using Strep-Tactin beads. **c)** Immunoblot showing TRIM5α down-regulation in HEK293T cells during RPXV and elephantpox virus infections. Cells were infected at 5 PFU/cell for the indicated times. MG132 was added at 2 hpi. In **b)**, cells were treated with 150 ng/mL doxycycline for 16 h before harvest. Inputs and AP proteins in **b)** or protein extracts in **c)** were analysed by SDS-PAGE and immunoblotted for epitope/protein. α-Actin, and C6 and D8 were controls for loading and viral infection, respectively. Data shown are representative of two independent experiments.

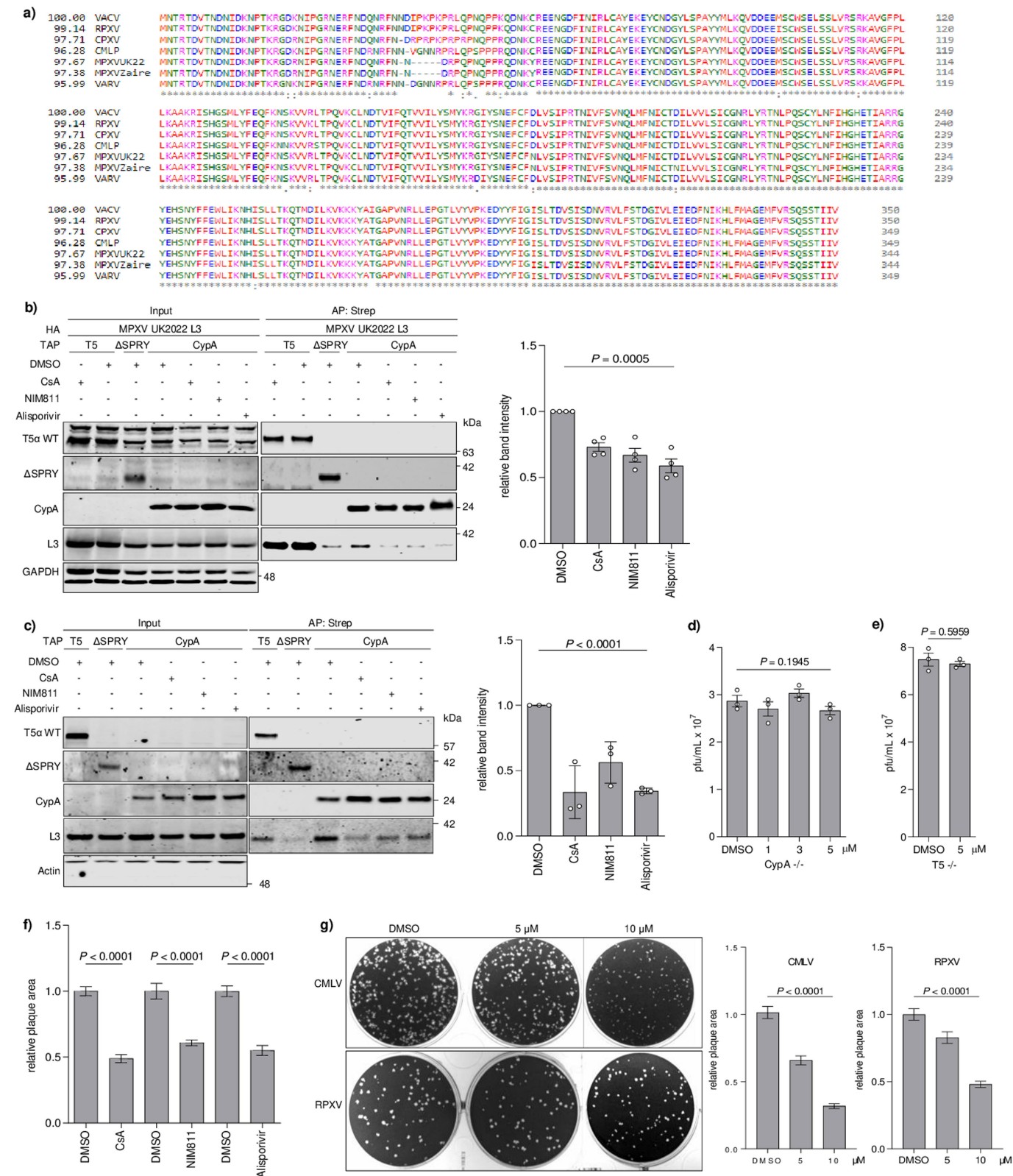

**Extended Data Fig. 8** | See next page for caption.

**Extended Data Fig. 8 | Conservation of L3 in VARV and MPXV and the effect of CsA and derivatives. a)** Amino acid sequence alignment of L3 proteins from orthopoxviruses created as in Extended Data Fig. 7a. **b)** Effect of CsA and derivatives, NIM811 or alisporivir, on MPXV UK2022 L3 co-precipitation with CypA. T-REx-293 TRIM5$^{-/-}$CypA$^{-/-}$ cells were co-transfected with HA-tagged L3 and TAP-tagged TRIM5α, ΔSPRY or CypA +/− 20 μM CsA, NIM811 or alisporivir 20 h before harvest. Graph shows band intensity of HA-tagged MPXV L3 in the rightmost 4 lanes from three independent experiments, normalised to TAP-tagged CypA and relative to the DMSO carrier sample. **c)** Effect of CsA and derivatives, NIM811 and alisporivir on endogenous VACV L3 co-precipitation with CypA. T-REx-293 TRIM5$^{-/-}$CypA$^{-/-}$ cells were co-transfected with TAP-tagged TRIM5α, ΔSPRY or CypA and infected with vΔC6 at 3 PFU/cell +/− 10 μM CsA, NIM811 or alisporivir for 12 h. The graph shows band intensity of VACV L3 from the rightmost 4 lanes from three independent experiments, normalised to TAP-tagged CypA and relative to the DMSO carrier sample. **d)** Effect of CsA on virus titres following VACV infection in T-REx-293 CypA$^{-/-}$ cells at 5 PFU/cell for 16 h.

n = 3/condition. **e)** Effect of CsA on virus titres following VACV infection in T-REx-293 TRIM5$^{-/-}$ cells at 5 PFU/cell for 16 h. n = 3/condition. **f)** Plaque area quantification of MPXV-CVR-S1 infection +/− 5 μM CsA and derivatives. BS-C-1 cells were infected with MPXV-CVR-S1 and 5 μM CsA, NIM811 or alisporivir were added to the semi-solid overlay. Plaques were measured using ImageJ and area relative to DMSO carrier were calculated. n ≥ 19/condition. **g)** Effect of CsA on plaque size following RPXV and CMLV infection in BS-C-1 cells. CsA at the indicated concentrations were added at 1 hpi. Plaques were measured using ImageJ and their area relative to DMSO carrier were calculated. n = 69/condition for CMLV and n = 68/condition for RPXV. Inputs and AP proteins in **b)** and **c)** were analysed by SDS-PAGE and immunoblotted for the indicated epitope/protein. Data shown in **b)** and **f)** is representative of four independent experiments, **c)** is from three and **d)**, **e)** and **g)** are from two. Data from **b)**, **c)**, **d)** and **g)** were analysed using One-Way Welch's ANOVA test. Data from **e)** and **f)** was analysed using two-tailed unpaired Student's t-test. Analyses were performed on GraphPad Prism. Mean ± s.e.m.

Jonas D. Albarnaz

# Reporting Summary

## Statistics

For all statistical analyses, confirm that the following items are present in the figure legend, table legend, main text, or Methods section.

| n/a | Confirmed | |
|---|---|---|
| ☐ | ☒ | The exact sample size (*n*) for each experimental group/condition, given as a discrete number and unit of measurement |
| ☐ | ☒ | A statement on whether measurements were taken from distinct samples or whether the same sample was measured repeatedly |
| ☐ | ☒ | The statistical test(s) used AND whether they are one- or two-sided *Only common tests should be described solely by name; describe more complex techniques in the Methods section.* |
| ☐ | ☒ | A description of all covariates tested |
| ☐ | ☒ | A description of any assumptions or corrections, such as tests of normality and adjustment for multiple comparisons |
| ☐ | ☒ | A full description of the statistical parameters including central tendency (e.g. means) or other basic estimates (e.g. regression coefficient) AND variation (e.g. standard deviation) or associated estimates of uncertainty (e.g. confidence intervals) |
| ☐ | ☒ | For null hypothesis testing, the test statistic (e.g. *F*, *t*, *r*) with confidence intervals, effect sizes, degrees of freedom and *P* value noted *Give P values as exact values whenever suitable.* |
| ☒ | ☐ | For Bayesian analysis, information on the choice of priors and Markov chain Monte Carlo settings |
| ☒ | ☐ | For hierarchical and complex designs, identification of the appropriate level for tests and full reporting of outcomes |
| ☒ | ☐ | Estimates of effect sizes (e.g. Cohen's *d*, Pearson's *r*), indicating how they were calculated |

*Our web collection on statistics for biologists contains articles on many of the points above.*

## Software and code

Policy information about availability of computer code

| Data collection | Image Studio Acquisition Software (version 5.2) ZEN Microscope Software (version 6.0.0.485) AxioVision (version 4.8) FLUOstar Omega Reader Control Software (version 1.20) MaxQuant (version 2.0.1.0) QuantStudioTM Real-Time software (v1.3) |
|---|---|
| Data analysis | GraphPad Prism (version 7.04) ZEN Lite Microscope Software (version 2.5.75.0) ImageJ-Fiji (version 1.53) Perseus (version 1.6.2.1) MARS Data Analysis Software (version 2.00) Image Studio Lite Quantification Software (version 5.5.4) Clustal Omega (1.20) |

For manuscripts utilizing custom algorithms or software that are central to the research but not yet described in published literature, software must be made available to editors and reviewers. We strongly encourage code deposition in a community repository (e.g. GitHub). See the Nature Portfolio guidelines for submitting code & software for further information.

## Data

Policy information about <u>availability of data</u>

All manuscripts must include a <u>data availability statement</u>. This statement should provide the following information, where applicable:

- Accession codes, unique identifiers, or web links for publicly available datasets
- A description of any restrictions on data availability
- For clinical datasets or third party data, please ensure that the statement adheres to our <u>policy</u>

All data from this study including supplementary material are available.

Proteomic data generated from label-free MS are uploaded in iProX repository, with the dataset identifier IPX0005650001.

Orthopoxvirus nucleotide sequences cited in this study are available on NCBI GenBank: VACV (YP_232972.1 and YP_232904.1), RPXV (AAS49792.1 and AAS49727.1), CPXV (ADZ24099.1 and NP_619819.1), CMLV (NP_570478.1 and NP_570410.1), MPXV UK 2022 (UWM73237.1 and UWM73173.1), MPXV Zaire (NP_536509.1 and NP_536509.1) and VARV major strain India 1967 (APR62813.1 and P0DSX3.1).

## Human research participants

Policy information about <u>studies involving human research participants and Sex and Gender in Research.</u>

| | |
|---|---|
| Reporting on sex and gender | N/A |
| Population characteristics | N/A |
| Recruitment | N/A |
| Ethics oversight | N/A |

Note that full information on the approval of the study protocol must also be provided in the manuscript.

# Field-specific reporting

Please select the one below that is the best fit for your research. If you are not sure, read the appropriate sections before making your selection.

☒ Life sciences ☐ Behavioural & social sciences ☐ Ecological, evolutionary & environmental sciences

For a reference copy of the document with all sections, see nature.com/documents/nr-reporting-summary-flat.pdf

# Life sciences study design

All studies must disclose on these points even when the disclosure is negative.

| | |
|---|---|
| Sample size | Base on our previous extensive experience studying orthopoxviruses, we selected a sufficiently large sample size for each study as follows:<br><br>Virus replication assays were carried out with n = 3 per condition<br>Virus spread assays were carried out with n ≥ 3 per condition<br>Virus plaque size measurements were recorded with n ≥ 6<br>Reporter gene assays were performed with n = 3 per condition, and 3 separate experiments<br>Immunoblot quantification were from 3 independent experiments<br>Immunofluorescence quantification was from n ≥ 36 per condition<br>MS was performed with n = 2 per condition<br>RT-qPCR was performed with n = 3 per condition, and 3 separate experiments |
| Data exclusions | No data exclusions in this study. |
| Replication | Virus replication and spread assays were performed twice, and virus plaque size measurement assays were performed at least three times. Graphs shown are consistent between independent experiments. Reporter gene assays are representative of three independent experiments. Co-precipitation and immunoblotting assays were performed three times and wheat germ transcription/translation assays were performed twice. Blots shown are representative of all repeats. Reporter gene assays are representative of three independent experiments. Immunofluorescence experiments were performed three times. Images shown are representative of all independent experiments. Mass spectrometry was performed once.<br>All repeat experiments gave the same result. |
| Randomization | For plaque size quantification, the plaques were picked randomly at each condition. |

Randomization is not applicable in the remaining experiments. Cell lines were allocated by genotype and were cultured side by side to minimize unpredicted environmental variations.

Blinding

Blinding was not applicable because this study does not include clinical trials. Researchers were not blinded to different cell lines. Virus infection experiments were conducted independently by Y. Zhao, Y. Lu and S. Richardson, whenever possible.

# Reporting for specific materials, systems and methods

We require information from authors about some types of materials, experimental systems and methods used in many studies. Here, indicate whether each material, system or method listed is relevant to your study. If you are not sure if a list item applies to your research, read the appropriate section before selecting a response.

## Materials & experimental systems

| n/a | Involved in the study |
|---|---|
| ☐ | ☒ Antibodies |
| ☐ | ☒ Eukaryotic cell lines |
| ☒ | ☐ Palaeontology and archaeology |
| ☒ | ☐ Animals and other organisms |
| ☒ | ☐ Clinical data |
| ☒ | ☐ Dual use research of concern |

## Methods

| n/a | Involved in the study |
|---|---|
| ☒ | ☐ ChIP-seq |
| ☒ | ☐ Flow cytometry |
| ☒ | ☐ MRI-based neuroimaging |

## Antibodies

Antibodies used

Mouse anti-FLAG Sigma-Aldrich Cat# F3165; RRID:AB_259529, dilution 1:1000
Rabbit anti-FLAG Sigma-Aldrich Cat# F7425; RRID:AB_439687, dilution 1:1000
Mouse anti-HA BioLegend Cat# 901513; RRID:AB_2565335, dilution 1:1000
Rabbit anti-HA Sigma-Aldrich Cat# H6908; RRID:AB_260070, dilution 1:1000
Mouse anti-Myc Merck Millipore Cat# 05-724; RRID:AB_11211891, dilution 1:1000
Mouse anti-GAPDH Sigma-Aldrich Cat# G8795; RRID:AB_1078991, dilution 1:1000
Rabbit anti-α-actin Sigma-Aldrich Cat# A2066; RRID:AB_476693, dilution 1:1000
Mouse anti-TRIM5 Santa Cruz Biotechnology Cat# sc-373864; RRID:AB_10918111, dilution 1:1000
Mouse anti-HDAC5 Santa Cruz Biotechnology Cat# sc-133225; RRID:AB_2116791, dilution 1:1000
Mouse anti-IKKβ Merck Millipore Cat# 05-535, RRID:AB_2122161, dilution 1:1000
Rabbit anti-CypA Invitrogen Cat# PA1-025; RRID:AB_2169124, dilution 1:1000
Rabbit anti-p-TAK1 Cell Signaling, Cat# 4508S; RRID:AB_561317, dilution 1:500
Rabbit anti-IkBa Cell Signaling, Cat# 9242S; RRID:AB_331623, dilution 1:1000
IRDye 680RD-conjugated goat anti-rabbit IgG LI-COR Cat# 926-68071; RRID:AB_10956166, dilution 1:1000
IRDye 800CW-conjugated goat anti-mouse IgG LI-COR Cat# 926-32210; RRID:AB_621842, dilution 1:1000
Donkey anti-mouse IgG (H+L) secondary antibody, Alexa Fluor 546 Invitrogen Cat# A10036; RRID:AB_2534012, dilution 1:5000
Goat anti-rabbit IgG (H+L) secondary antibody, Alexa Fluor 488 Invitrogen Cat# A11008; RRID:AB_143165, dilution 1:5000
Donkey anti-rabbit IgG (H+L) secondary antibody, Alexa Fluor 546 Invitrogen Cat# A11010; RRID:AB_2534077, dilution 1:5000
Rabbit anti-C6 Unterholzner et al., 2011 N/A, dilution 1:1000
Mouse anti-D8 Parkinson & Smith, 1994  N/A, dilution 1:1000
Rabbit anti-L3 Resch & Moss, 2005 N/A, dilution 1:1000
Rabbit anti-C6 and Mouse anti-D8 were produced in this lab.
Rabbit anti-L3 was a kind gift from Dr Bernard Moss, NIAID.

Validation

The Rabbit anti-C6 antibody was validated in Unterholzner et al., 2011, the Rabbit anti-L3 in Resch & Moss, 2005, the Mouse anti-D8 in Parkinson & Smith, 1994, and the Mouse anti-HDAC5 in Soday et al., 2019, (Fig. 7D, 7H). The Mouse anti-TRIM5 (Fig. ED 2a, 2b) and the Rabbit anti-CypA (Fig. ED 3a, 3b) antibodies were validated in this study using knockout cell lines lacking TRIM5 or CypA.

Validation detail for the other commercial antibodies is available on the manufacturer's website that is attached as following.
Mouse anti-FLAG Sigma-Aldrich Cat# F3165; https://www.sigmaaldrich.com/GB/en/product/sigma/f3165
Rabbit anti-FLAG Sigma-Aldrich Cat# F7425; https://www.sigmaaldrich.com/GB/en/product/sigma/f7425
Mouse anti-HA BioLegend Cat# 901513; https://www.biolegend.com/en-us/products/anti-ha-11-epitope-tag-antibody-11071?GroupID=GROUP26
Rabbit anti-HA Sigma-Aldrich Cat# H6908; https://www.sigmaaldrich.com/GB/en/product/sigma/h6908
Mouse anti-Myc Merck Millipore Cat# 05-724; https://www.merckmillipore.com/GB/en/product/Anti-Myc-Tag-Antibody-clone-4A6,MM_NF-05-724
Mouse anti-GAPDH Sigma-Aldrich Cat# G8795; https://www.sigmaaldrich.com/GB/en/product/sigma/g8795
Rabbit anti-α-actin Sigma-Aldrich Cat# A2066; https://www.sigmaaldrich.com/GB/en/product/sigma/a2066
Rabbit anti-p-TAK1 Cell Signaling, Cat# 4508S; https://www.cellsignal.com/products/primary-antibodies/phospho-tak1-thr184-187-90c7-rabbit-mab/4508
Rabbit anti-IkBa Cell Signaling, Cat# 9242S; https://www.cellsignal.com/products/primary-antibodies/ikba-antibody/9242
IRDye 680RD-conjugated goat anti-rabbit IgG LI-COR Cat# 926-68071; https://www.licor.com/bio/reagents/irdye-680rd-goat-anti-rabbit-igg-secondary-antibody
IRDye 800CW-conjugated goat anti-mouse IgG LI-COR Cat# 926-32210; https://www.licor.com/bio/reagents/irdye-800cw-goat-anti-mouse-igg-secondary-antibody
Donkey anti-mouse IgG (H+L) secondary antibody, Alexa Fluor 546 Invitrogen Cat# A10036; https://www.thermofisher.com/

antibody/product/Donkey-anti-Mouse-IgG-H-L-Highly-Cross-Adsorbed-Secondary-Antibody-Polyclonal/A10036
Goat anti-rabbit IgG (H+L) secondary antibody, Alexa Fluor 488 Invitrogen Cat# A11008; https://www.thermofisher.com/antibody/product/Goat-anti-Rabbit-IgG-H-L-Cross-Adsorbed-Secondary-Antibody-Polyclonal/A-11008
Goat anti-rabbit IgG (H+L) secondary antibody, Alexa Fluor 546 Invitrogen Cat# A11010; https://www.thermofisher.com/antibody/product/Goat-anti-Rabbit-IgG-H-L-Cross-Adsorbed-Secondary-Antibody-Polyclonal/A-11010

# Eukaryotic cell lines

Policy information about cell lines and Sex and Gender in Research

| Cell line source(s) | Human foetal foreskin fibroblasts (HFFFs) immortalized with human telomerase (HFFF-TERTs) was a kind gift from Prof Michael Weekes (University of Cambridge), who obtained them from Prof Richard Stanton (Cardiff University), PMID: 17522202.<br>BS-C-1 (African green monkey cell line) ATCC, ATCC: CCL-26<br>RK13 cells (rabbit kidney cell line) ATCC, ATCC: CCL-37<br>HeLa (human cervical adenocarcinoma epithelial cell line) ATCC, ATCC: CCL-2<br>HEK293T (human embryo kidney epithelial cell line) ATCC, ATCC: CRL-11268<br>T-REx-293 Life technologies, R71007<br>CRISPR Ctrl HeLa (for TRIM5-/-) This paper<br>TRIM5-/- clone 1 HeLa This paper<br>TRIM5-/- clone 2 HeLa This paper<br>CRISPR RICE T-REx-293 (for TRIM5-/-) This paper<br>TRIM5-/- clone 1 T-REx-293 This paper<br>TRIM5-/- clone 2 T-REx-293 This paper<br>EV T-REx-293 in T5-/- (complementation) This paper<br>TAP-T5α WT T-REx-293 (complementation) This paper<br>TAP-T5γ WT T-REx-293 (complementation) This paper<br>TAP-T5δ WT T-REx-293 (complementation) This paper<br>EV T-REx-293 (over-expression) This paper<br>FLAG-T5α WT T-REx-293 (over-expression) This paper<br>FLAG-T5γ WT T-REx-293 (over-expression) This paper<br>FLAG-T5δ WT T-REx-293 (over-expression) This paper<br>FLAG-T5α L19R T-REx-293 (complementation) This paper<br>FLAG-T5α N70A T-REx-293 (complementation) This paper<br>FLAG-T5α R119E T-REx-293 (complementation) This paper<br>FLAG-T5α ΔSPRY T-REx-293 (complementation) This paper<br>CRISPR/Cas9 control T-REx-293 (for CypA-/-) This paper<br>CypA-/- clone 1 T-REx-293 This paper<br>CypA-/- clone 2 T-REx-293 This paper<br>EV T-REx-293 in CypA-/- (complementation) This paper<br>TAP-CypA WT T-REx-293 (complementation) This paper<br>TAP-CypA R55A T-REx-293 (complementation) This paper<br>TAP-CypA F113A T-REx-293 (complementation) This paper<br>CRISPR RICE T-REx-293 (for TRIM5-/- CypA-/-) This paper<br>T5-/- CypA-/- clone 1 T-REx-293 This paper<br>T5-/- CypA-/- clone 2 T-REx-293 This paper<br>EV T5-/- CypA-/- T-REx-293 (complementation) This paper<br>TAP-CypA T5-/- CypA-/- T-REx-293 (complementation) This paper<br>Myc-VACVC6 HEK293T (over-expression) Soday et al., 2019<br>TAP-VACVB14 HEK293T (over-expression) Lu et al., 2019<br>TAP-VACVcoC6 HEK293T (over-expression) Lu et al., 2019<br>TAP-MPXVUK22B14 T-REx-293 (over-expression) This paper<br>TAP-VACVcoC6 T-REx-293 (over-expression) This paper<br>TAP-RPXVC6 T-REx-293 (over-expression) This paper<br>TAP-CPXVC6 T-REx-293 (over-expression) This paper<br>TAP-ElephantpoxvirusC6 T-REx-293 (over-expression) This paper<br>TAP-CMLVC6 T-REx-293 (over-expression) This paper<br>TAP-MPXVUK2022coC6 T-REx-293 (over-expression) This paper<br>TAP-MPXVZairecoC6 T-REx-293 (over-expression) This paper<br>TAP-VARVcoC6 T-REx-293 (over-expression) This paper<br>TAP-TAPcoL3 T-REx-293 (over-expression) This paper |
| --- | --- |
| Authentication | Commercial cell lines were authenticated by the suppliers, detail of the authentication is provided as following,<br>BS-C-1 (African green monkey cell line) ATCC: CCL-26, https://www.atcc.org/products/ccl-26<br>RK13 cells (rabbit kidney cell line) ATCC: CCL-37, https://www.atcc.org/products/ccl-37<br>HeLa (human cervical adenocarcinoma epithelial cell line) ATCC: CCL-2, https://www.atcc.org/products/ccl-2<br>HEK293T (human embryo kidney epithelial cell line) ATCC: CRL-11268, https://www.atcc.org/products/crl-11268<br>The above cell lines were authenticated by ATCC,  and a link to each cell line describing the authentication is attached.<br>T-REx-293 Life technologies, R71007 were authenticated by Life technologies. https://www.thermofisher.com/order/catalog/product/R71007<br><br>HFFFs were tested regularly to confirm that human leukocyte antigen (HLA) and MHC Class I Polypeptide-Related Sequence A (MICA) genotypes, cell morphology and antibiotic resistance are unchanged. In addition, HCMV only replicates in human fibroblast cells (dermal or foreskin in origin) and so HCMV infection was tested and confirmed in HFFF cells. |

Mycoplasma contamination | The authors confirm that all cell lines tested negative for mycoplasma contamination.

Commonly misidentified lines
(See ICLAC register) | No commonly misidentified cell lines were used in this study.

