## [Peer Review File · Nature]

Manuscript Title: TRIM5 α restricts poxviruses and is antagonised by CypA and viral protein C6

Reviewer Comments & Author Rebuttals

Reviewer Reports on the Initial Version:

Referees' comments:

Referee #1 (Remarks to the Author):

This manuscript makes a compelling case that human TRIM5 α can act a poxvirus restriction factor following binding to capsid protein L3, diminishing virus replication and activating innate immunity. This observation is novel and interesting. I consider the work of great potential interest in particular regarding the idea that specific host proteins can recognize and restrict several entirely different kinds of virus though it might be nice to learn more about the mechanisms by which TRIM5 α can recognize such diverse targets.

Based on the analysis of viral strategies to evade restriction, novel approaches to combatting orthopoxviruses such as monkeypox are suggested. However, I do have reservations as to whether the relatively limited apparent effects on virus titre reported here will provide the basis for new therapies. Is it likely that the levels inhibition observed here will have in vivo benefits? How do they compare with the effects of tecovirimat? On other kinds of virus? In addition, it seems important to note that while CypA may protect HIV against human TRIM5 α , it does not seem to have an impact on restriction by other TRIM5 α s, so it is unwise to draw general conclusions about its role in restriction. Given that human TRIM5 α is unique in this respect, speculations about fine specificity may also be unwarranted. It would be interesting to test other primate TRIM5 α for restriction of different poxviruses.

Although I have considerable enthusiasm for this report, I did find it rather hard to read. There is an enormous amount of data here but with relatively little detail. It would appear that the legends to the figures are simply lists; more detail would greatly help the reader. Some additional background/narrative would be helpful for many particularly those unfamiliar with poxvirus biology and the various techniques used here. For example, it might have been helpful (i) to say more about the roles of L3 and C6 in poxvirus replication (to better appreciate similarities and differences between restriction of different kinds of virus) and (ii) to explain the quantitation of TRIM5 α in cytoplasm versus virus factories (won't virus associated decreases in protein level skew scoring punctate v cytoplasmic expression?).

Referee #2 (Remarks to the Author):

Orthopoxviruses encode many proteins that are used for defense against cellular innate immunity. In this study, Zhong and co-workers show that the cellular protein TRIM5a is an orthopoxvirus restriction factor, then demonstrate that the orthopoxviral protein C6 binds TRIM5a and induces proteosomal degradation by an unknown mechanism. (Interestingly C6 was previously shown to bind unrelated cell proteins and inhibit IRF3 activation). Furthermore, another orthopoxviral protein L3 recruits cyclophilin A, a cellular protein that antagonizes TRIM5a, to sites of virus replication where it can provide a second level of defense. Finally, they show that two drugs that are known to target cyclophilin have anti-viral properties. The findings are novel and significant, the experiments are well designed, and the paper is clearly written. Taken together, these data make a nice story of host defense and orthopoxvirus counter defense.

Comments

1. Evidence that L3 induces NF- κ B activation is based on transfection. However, L3 is made late in

infection. Do the authors think that virion-associated L3 can mediate NF-KB activation? Could this be tested?

2. How does binding of TRIM5a to L3 at late times of infection inhibit virus replication? Does this interaction prevent packaging of L3? Is TRIM5a bound to L3 incorporated into virus particles?

3. The authors used the Welch t-test for all statistical analysis, which mostly consists of only 3 data points. Could the data from the repeat experiments be combined and/or have the repeat experiments also been analyzed for significance?

4. While the Welch t-test is valid for experiments in which 2 means are compared, the Welch t-test was also used here for comparing multiple means e.g. Fig. 2B, 2C; Fig. 3B, 3C, 3E, 3F, 3G, Fig. 4H, 4I; Fig. 6C, 6E for which ANOVA or another multiple means test would be more appropriate.

Bernard Moss

Referee #3 (Remarks to the Author):

This is an exciting paper that reveals for the first time that TRIM5alpha (T5a) is a restriction factor for orthopoxviruses. Up to now, T5a was only known to restrict RNA viruses, especially HIV-1. The authors take a broad approach to implicate and explore T5a as an anti-orthopoxvirus restriction factor, including identifying the viral target for T5a as L3, a conserved poxviral structural protein. They also reveal a viral antagonist of T5a, namely C6, a known virulence factor also highly conserved across poxviruses. They also show that the cellular protein CypA antagonises T5a restriction activity against L3, and thus has a proviral effect, which can be antagonised by cyclosporin A derivatives proposed previously to be anti-viral. Such derivatives are confirmed here to have anti-poxviral activity and thus represent a potential host-directed anti-viral therapy for poxviral diseases such as mpox.

The strengths of the paper are the novelty of the results, and the broad approach taken to identify both a new anti-poxviral host restriction pathway, and poxviral antagonists of such a pathway. The broad approach has some weaknesses in not going deep enough to clarify some mechanistic aspects of the system, these are outlined below together with some other, mainly minor, issues that should be addressed:

1. In Fig 1b and d why does MG132 treatment only partially restore T5a protein levels, if T5a degradation is mediated by the proteasome?
2. In introducing N1 please explain the rationale for its use as a control poxviral protein. It is only later in the paper briefly referred to that N1 (and C6) are Bcl-2-like proteins.
3. C6 is identified as the first viral protein to degrade T5a, via a proteasomal pathway. If this is the case then C6 expression would be expected to cause the appearance of K48 ubiquitin chains on T5a. Is this so?
4. In Fig 4a could other poxviral proteins identified as interacting with T5a be targets for restriction as well as L3? Please discuss.
5. The mechanism whereby T5a restricts poxviruses via L3 binding is not clear, especially since the authors cite that viruses lacking L3 still form intact virions (ref 28). This suggests a different restriction mechanism than disruption of virions by oligomers of T5a binding, as is the case for retroviruses. As well as showing localisation of T5a to viral factories, is it possible to visualise T5a bound to poxviral virions? Is there direct evidence that T5a binds to an intact virion to disrupt it?
6. The confocal images from Fig S4 would be better placed in the main Figures.
7. The section on activation by NFkappaB by T5a and L3 is very under-developed, consisting only of a reporter gene assay analysis of NFkappaB. Activation of innate immunity by T5a in the context of a poxviral infection, and by L3 is an important aspect of the study. It would be quite straightforward for the authors to assess NFkappaB activation with alternative assays, as they

have used in their previous papers, in order to gain more insights into how T5a and L3 stimulate NFkappaB gene promoter activation, and to be more convinced of the conclusion (e.g. IkappaB degradation, p65 phosphorylation, p65 nuclear translocation).

8. Related to (7), how does T5a activate NFkappaB in the context of a poxviral infection? IS it via polyubiquitination of TAK1, as has been shown previously for HIV-1 infection?

Author Rebuttals to Initial Comments:

Referee #1 (Remarks to the Author):

This manuscript makes a compelling case that human TRIM5 α can act a poxvirus restriction factor following binding to capsid protein L3, diminishing virus replication and activating innate immunity. This observation is novel and interesting. I consider the work of great potential interest in particular regarding the idea that specific host proteins can recognize and restrict several entirely different kinds of virus though it might be nice to learn more about the mechanisms by which TRIM5 α can recognize such diverse targets.

We thank the reviewer for a positive summary of the paper. Concerning the mechanism of TRIM5 α -mediated restriction of VACV, we have added a new Figure (Fig. S4) that explores further the observation that protein L3 can dimerise and shows that WT TRIM5 α , but not TRIM5 α lacking Ub ligase activity, promotes dimerisation and WT cyclophilin A (CypA), but not enzymatically inactive CypA, antagonises dimerisation (Fig. S4a-d). We also show that L3 is modified biochemically by TRIM5 α , with a ladder of higher molecular mass proteins that are highly reminiscent of ubiquitylation (Fig. S4e). These modifications are not induced by enzymatically inactive TRIM5 α and are reversed by addition of CypA (Fig. S4e). Since enzymatically active TRIM5 α is needed for restriction of VACV, it implies that post-translation modification (likely ubiquitylation) of L3 is also needed for restriction.

So unlike the situation with retroviruses where the recognition of the retrovirus capsid by TRIM5 α causes its premature disassembly and degradation, with poxviruses, the dimerisation of L3 is enhanced by TRIM5 α and this is antagonised by CypA. How the modification and enhanced dimerization of L3 leads to virus restriction, and how the SPRY domain of TRIM5 α can recognise such diverse proteins, are the topics of future study. One possibility to explain the recognition of these diverse proteins is their dimeric or oligomeric nature.

Based on the analysis of viral strategies to evade restriction, novel approaches to combatting orthopoxviruses such as monkeypox are suggested. However, I do have reservations as to whether the relatively limited apparent effects on virus titre reported here will provide the basis for new therapies. Is it likely that the levels inhibition observed here will have *in vivo* benefits? How do they compare with the effects of tecovirimat? On other kinds of virus?

We agree with the reviewer that the changes in viral titre are modest, but such a change as well as the reduction in viral spread, as indicated by reduced plaque size, can have a big impact *in vivo*. Note that the drug tecovirimat does not inhibit virus replication at all, rather it blocks formation of extracellular virus and thereby virus spread and so greatly diminishes plaque size. Yet tecovirimat is an effective drug against orthopoxviruses *in vivo* – so long as the infected host has an effective immune system to clear virus-infected cells. Tecovirimat is ineffective in immunocompromised patients, as illustrated by a fatal case of cowpox in a kidney transplant patient (see Wendt et al. *Int J Infect Dis* 106: 276-278, 2021).

It seems to us that, given it is known that tecovirimat does not inhibit virus replication, there may not be a great benefit in comparing its ability to inhibit replication with CsA or derivatives. It will not inhibit, whereas CsA will. Equally, the effect of tecovirimat on virus plaque formation is known: it causes a profound reduction in plaque size that is equivalent to the reduction in plaque size caused by deletion of the *F13L* gene, which encodes the F13 protein that is the target of tecovirimat. So little might be gained by comparing this with CsA and derivatives.

In terms of the effect of CsA on disease outcome *in vivo*, this will need experimental determination in the future of course, but it is quite possible that it may attenuate disease severity substantially. This is because mutations that diminish the size of plaques formed by VACV, even if they give no alteration in infectious virus titre, will have profound effects on virus virulence. See the consequence of deleting either the *A36R* or *F12L* genes for instance. Neither encoded protein is needed for virus replication, but both are needed for spread and viruses lacking them are highly attenuated in mouse models of infection (see Parkinson & Smith *Virology* 204, 376, 1994; Zhang et al., *J Virol* 74, 11663, 2000). Whereas WT and revertant viruses cause disease in mice at a given dose, the F12 deletion mutant is completely avirulent at the same dose. And even if that dose is increased 10,000 fold, the deletion mutant is still avirulent.

With regards to “other kinds of virus”, we have shown that the drugs diminish the yield of infectious virus and plaque size with both VACV and monkeypox virus (MPXV). Further, we have shown that the viral target for TRIM5 α and CypA, the capsid protein L3, is highly conserved across the orthopoxviruses and L3 protein expressed ectopically from VACV, MPXV and variola viruses all co-precipitate with TRIM5 α and CypA. In each case the interaction between CypA and L3 is inhibited by addition of CsA. These biochemical assays indicate that the drug is likely to also work against these other viruses, and we have now shown this for two other orthopoxviruses, rabbitpox and camelpox virus (closely related to VARV), and these additional data are included in Fig. S6f.

In addition, it seems important to note that while CypA may protect HIV against human TRIM5 α , it does not seem to have an impact on restriction by other TRIM5 α s, so it is unwise to draw general conclusions about its role in restriction. Given that human TRIM5 α is unique in this respect, speculations about fine specificity may also be unwarranted. It would be interesting to test other primate TRIM5 α for restriction of different poxviruses.

We agree with the reviewer that one cannot be sure that TRIM5 α from different species will have the same effect. However, we have provided data with multiple cell types, showing that in human cells, the titre of virus is reduced by CsA, and in African green monkey (*Cercopithecus aethiops*) kidney cells, the size of plaques is reduced by CsA. The latter indicates that this non-human primate TRIM5 α can restrict orthopoxviruses. Furthermore, TRIM5 α and CypA bind to L3 derived from orthopoxviruses adapted to different species, so it is clear already that this will not be a case of the drug only working against a human CypA in human cells with a human virus. This is a very interesting area and will certainly need further experimentation in the future.

Although I have considerable enthusiasm for this report, I did find it rather hard to read. There is an enormous amount of data here but with relatively little detail. It would appear that the legends to the figures are simply lists; more detail would greatly help the reader.

We agree with the reviewer and do apologise for the brief format of the figure legends, but *Nature* allows very few words in the legends and we have had to precis the legends considerably to approach or meet this word limit. If *Nature* would permit a greater word limit for the legends we would be happy to expand them to help the reviewer and readers.

Some additional background/narrative would be helpful for many particularly those unfamiliar with poxvirus biology and the various techniques used here. For example, it might have been helpful (i) to say more about the roles of L3 and C6 in poxvirus replication (to better appreciate similarities and differences between restriction of different kinds of virus) and (ii) to explain the quantitation of TRIM5 α in cytoplasm versus virus factories (won't virus associated decreases in protein level skew scoring punctate v cytoplasmic expression?).

For the C6 and L3 proteins, we did provide a short summary of what is known about them, see lines 185-8 and 213-8 of the original submission. C6 is non-essential for virus replication and is predicted to have a Bcl-2-like structure, and has several roles in evasion of innate immunity, such as blocking IRF3 activation and restricting JAK-STAT signalling downstream of type I IFN by inducing degradation of HDAC4 and thereby reducing STAT2 recruitment to ISRE bearing promoters. In contrast, little is known about L3 except that it is essential for the production of infectious virions. There is no structure known for L3, and AlphaFold did not predict a structure that is similar to retrovirus capsids. Instead, it predicted that VACV L3 might be derived from an ancestral serine-threonine protein kinase but now lacks key residues needed for enzymatic activity (Mutz et al. mBio e0040823, 2023). Nevertheless, like the retrovirus capsid protein, we have demonstrated that L3 can dimerise, and this is enhanced by TRIM5 α (Fig. S4).

Concerning the quantitation of the different location of TRIM5 α in the presence or absence of L3 (+/- IPTG), the number of cells in which TRIM5 α is either localised in the viral factory only, or also elsewhere in the cytoplasm was measured relative to the total number of cells per condition. Any virus-associated decrease in TRIM5 α will be common to all infections, because protein C6 will be expressed in all conditions, and therefore, there should be no skewing of scoring.

Referee #2 Bernard Moss (Remarks to the Author):

Orthopoxviruses encode many proteins that are used for defense against cellular innate immunity. In this study, Zhong and co-workers show that the cellular protein TRIM5a is an orthopoxvirus restriction factor, then demonstrate that the orthopoxviral protein C6 binds TRIM5a and induces proteosomal degradation by an unknown mechanism. (Interestingly C6 was previously shown to bind unrelated cell proteins and inhibit IRF3 activation). Furthermore, another orthopoxviral protein L3 recruits cyclophilin A, a cellular protein that antagonizes TRIM5a, to sites of virus replication where it can provide a second level of defense. Finally, they show that two drugs that are known to target cyclophilin have anti-viral properties. The findings are novel and significant, the experiments are well designed, and the paper is clearly written. Taken together, these data make a nice story of host defense and orthopoxvirus counter defense.

We thank Dr Moss for his positive summary of the paper.

Comments

1. Evidence that L3 induces NF- κ B activation is based on transfection. However, L3 is made late in infection. Do the authors think that virion-associated L3 can mediate NF- κ B activation? Could this be tested?

It is possible that L3 on the incoming virion might activate NF- κ B, although this may be technically very difficult to demonstrate. First, it would be necessary to prepare highly purified stocks of virus that do or do not contain L3 and then infect cells with these in the presence or absence of an inhibitor of virus gene expression, to see if NF- κ B activation can be detected in the presence or absence of L3. Whether this will be possible is unclear and may depend on whether or not there are additional virion structures that also activate NF- κ B,

whether contaminating material that activates the pathway can be eliminated sufficiently when virus is purified from the cytoplasm of infected cells, and whether the very many inhibitors of NF- κ B that VACV expresses will mask any signal detected from incoming virions. Given the technically challenging nature of these experiments and the uncertainty of them working, we would prioritise other experiments ahead of this, such as, for instance, seeking what additional host protein(s) L3 binds to in TRIM5^{-/-} cells to activate innate immunity. Or, put another way, what is the cellular sensor of L3 that activates NF- κ B?

2. How does binding of TRIM5 α to L3 at late times of infection inhibit virus replication? Does this interaction prevent packaging of L3? Is TRIM5 α bound to L3 incorporated into virus particles?

We agree that the detailed mechanism of how TRIM5 α restricts orthopoxvirus replication remains to be worked out, but feel we have gone quite a long way for a first report of a new phenomenon. Specifically, as well as demonstrating reduced plaque size and virus titres in the presence of TRIM5 α , we have undertaken a mutational analysis to identify which parts of TRIM5 α , and which enzymatic activity, is needed for virus restriction. We have identified the virus target protein, L3, mapped the region of TRIM5 α needed for this interaction and showed it is via direct binding. We have shown that the virus target is highly conserved amongst orthopoxviruses, and that the L3 protein from the important human pathogens, variola and monkeypox viruses, co-precipitate with TRIM5 α and CypA. Furthermore, the latter interaction is disrupted by CsA and related compounds. In a parallel with retroviruses, we have shown that the viral target of TRIM5 α protein, L3, is a capsid protein that can dimerise, is modified biochemically by TRIM5 α , and TRIM5 α enhances L3 dimerisation. The modification of L3 by TRIM5 α , is very reminiscent of ubiquitylation and, consistent with this, an analysis of ubiquitylated proteins from cowpox virus identified the K272 of the cowpox virus L3 orthologue as a site of ubiquitylation (Grossegesse et al., Sci Rep 8, 1807, 2018). This lysine is conserved in L3 from orthopoxviruses.

The paper also demonstrates that the L3 protein alone can activate innate immunity via NF- κ B signalling and that it synergises with TRIM5 α to give more potent activation. Further, this synergistic activation is reversed by CypA, but not by a catalytically inactive mutant of CypA that lacks prolyl isomerase activity.

Having said all that, we have added additional data (Fig. S4) that is described in detail in response to reviewer 1. In summary, we find that the dimerisation of L3 is greater in the presence of TRIM5 α and the ability of TRIM5 α to stabilise L3 dimers requires TRIM5 α E3 ubiquitin ligase activity, and the L3 protein is modified biochemically by this activity. Also, we show that the stabilisation of L3 dimers by TRIM5 α is reversed by CypA and that this requires the enzymatic activity of CypA, so providing a link between the anti-viral activity of TRIM5 α and the pro-viral activity of CypA.

To really work out mechanism and dissect the contribution of activation of innate immunity by L3/TRIM5 α versus modification of L3 via TRIM5 α is a big task and we wonder if this is needed for this first report.

3. The authors used the Welch t-test for all statistical analysis, which mostly consists of only 3 data points. Could the data from the repeat experiments be combined and/or have the repeat experiments also been analyzed for significance?

Repeat experiments, although not presented, were also analysed for significance and graphs shown in the figures are representative of all repeats. Although the conclusions drawn from the repeat experiments are consistent, the variation between independent experiments may mask significance if combined together.

4. While the Welch t-test is valid for experiments in which 2 means are compared, the Welch t-test was also used here for comparing multiple means e.g. Fig. 2B, 2C; Fig. 3B, 3C, 3E, 3F, 3G, Fig. 4H, 4I; Fig. 6C, 6E for which ANOVA or another multiple means test would be more appropriate.

In instances where ANOVA is applicable, we have modified the statistical analysis according to Dr Moss' recommendations (Figs 2b, 2c, S2a, S2c-e, 3b, 3c, 3e, 3f, S3a, S3b, 6e, 6f, S6b-d, S6f).

However, we do not feel that ANOVA is appropriate in every case. Specifically, we feel presenting a single P value for all the means would be unsuitable in cases where not all conditions show statistical significance in pairwise comparisons against the same control group, or where it is not logical to compare each condition with all other conditions.

For example, in Fig. 6c, which investigates the dose-dependent effect of CsA and derivatives on VACV infectious titre after high MOI, 1 μM alisporivir treatment did not yield a significant difference compared to the DMSO control, whereas the higher concentrations did. This change in significance is noted when each of the concentrations were compared to DMSO in a pairwise manner (using t-test). However, with ANOVA, although the difference between DMSO and alisporivir-treated samples remain significant ($P = 0.0108$), we feel it overlooks the difference in significance between DMSO vs 1 μM alisporivir and DMSO vs 3/5 μM alisporivir and therefore, may not be able to fairly represent the data.

Fig. 6c. Effect of CsA and derivatives on infectious VACV titre following infection in T-REx-293 cells at 5 PFU/cell for 16 h. $n=3$ /condition. Data are representative of two independent experiments and were analysed using two-tailed unpaired Student's t-test on GraphPad Prism. Mean \pm s.e.m.

Another different situation is Fig. 4i (below) that investigates the activation of NF- κ B mediated by TRIM5 α and L3 and is reversed by CypA WT but not by the enzymatically defective mutant. Here, not all samples are comparable such as CypA vs L3+TRIM5 α or TRIM5 α vs L3+WT CypA. Using ANOVA in this case will unnecessarily skew where the pairwise comparison is applicable (indicated in figure).

Fig. 4i. TRIM5 α and L3 stimulate NF- κ B activation synergistically and this is reduced by WT CypA. T-REx-293 TRIM5 $^{-/-}$ CypA $^{-/-}$ cells were co-transfected with NF- κ B-luciferase and Renilla luciferase reporters alongside either EV, TRIM5 α , L3 or CypA WT individually, or TRIM5 α and L3 with CypA WT or mutant, R55A, or L3 with CypA WT or R55A. Cells were harvested 20 h after transfection, firefly luciferase activity was measured and normalised to Renilla luciferase. Fold induction is relative to EV. n=3/condition.

Referee #3 (Remarks to the Author):

This is an exciting paper that reveals for the first time that TRIM5 α (T5a) is a restriction factor for orthopoxviruses. Up to now, T5a was only known to restrict RNA viruses, especially HIV-1. The authors take a broad approach to implicate and explore T5a as an anti-orthopoxvirus restriction factor, including identifying the viral target for T5a as L3, a conserved poxviral structural protein. They also reveal a viral antagonist of T5a, namely C6, a known virulence factor also highly conserved across poxviruses. They also show that the cellular protein CypA antagonises T5a restriction activity against L3, and thus has a proviral effect, which can be antagonised by cyclosporin A derivatives proposed previously to be anti-viral. Such derivatives are confirmed here to have anti-poxviral activity and thus represent a potential host-directed anti-viral therapy for poxviral diseases such as mpox.

We thank the reviewer for the positive summary of the paper.

The strengths of the paper are the novelty of the results, and the broad approach taken to identify both a new anti-poxviral host restriction pathway, and poxviral antagonists of such a pathway. The broad approach has some weaknesses in not going deep enough to clarify some mechanistic aspects of the system, these are outlined below together with some other, mainly minor, issues that should be addressed:

1. In Fig 1b and d why does MG132 treatment only partially restore T5a protein levels, if T5a degradation is mediated by the proteasome?

This is likely because of the VACV-induced inhibition of host protein synthesis (Moss, B. J Virol 2, 1068, 1968) so that a protein like TRIM5 α that has an appreciable rate of turnover naturally, cannot be re-synthesised during infection and so declines somewhat independent of C6-induced degradation.

2. In introducing N1 please explain the rationale for its use as a control poxviral protein. It is only later in the paper briefly referred to that N1 (and C6) are Bcl-2-like proteins.

As requested, we will explain why the protein N1 is selected when first mentioned, rather than later in the paper (it is another VACV Bcl-2 protein that regulates innate immunity).

3. C6 is identified as the first viral protein to degrade T5a, via a proteasomal pathway. If this is the case then C6 expression would be expected to cause the appearance of K48 ubiquitin chains on T5a. Is this so?

To address this issue, we have expressed tagged-WT Ub or tagged-Ub in which only K63 or K48 ubiquitylation is possible, alongside TRIM5 α N70A (so that no T5 autoubiquitylation occurs, Fletcher et al., Cell Host Microbe, 24, 761, 2018) and either protein C6 or another VACV Bcl-2 protein called B14. After precipitation of TRIM5 α via an HA tag, immunoblotting revealed higher molecular mass TRIM5 α bands (Fig. S1c). This shows that i) C6 enhances the ubiquitylation of TRIM5 α compared to B14, and ii) this is diminished by K48-Ub but not by K63-Ub (Fig. S1c). Notably, autoubiquitylation of TRIM5 α also occurs via K63 (Fletcher et al., EMBO J, 34, 2078, 2015).

4. In Fig 4a could other poxviral proteins identified as interacting with T5a be targets for restriction as well as L3? Please discuss.

We thank the reviewer for raising this point. We focussed on structural proteins and, in addition to L3, proteins J6, A24, D8, F13 and D1 were enriched in the TRIM5 α pulldown. But of these, only L3 was pulled down by both TRIM5 and CypA, and only with L3 was the interaction with CypA lost in the presence of CsA. These observations fitted our selection criteria so that is why we choose L3 for further analysis.

We agree that it is possible that one or more of these proteins are a target of TRIM5 α , while not interacting with CypA. J6 and A24 are subunits of the RNA polymerase, and D1 is a subunit of the capping enzyme. These proteins, being enzymes, are expressed at relatively low levels and all are associated within the virus core. F13 is not associated with mature virions (MVs), is not needed for their replication, and does not localise to the virus factories as TRIM5 α does. D8 is a MV transmembrane protein, and although immunofluorescence detects its localisation to the virus factories, it is not exclusively there like TRIM5 α , and does not show co-localisation with TRIM5 α puncta (below). Please note these data are to help with the review process rather than to include in the manuscript.

TRIM5 α localises to viral factories during infection

T-REx-293 TRIM5 $^{-/-}$ cells stably transfected with TAP-tagged TRIM5 α were induced with 150 ng/mL doxycycline to express tagged proteins 24 h prior to infection with v6/2 at 2.5 p.f.u./cell for 12 h. A mock-infected sample was also included. Cells were stained with DAPI (blue),

antibodies against FLAG-tagged TRIM5 α (red) and the late VACV protein D8 (green). White arrows indicate viral factories. Images were acquired using Zeiss LSM 700.

We do recognise that the interaction between TRIM5 α and one or more of these additional proteins may have biological relevance, but this would be independent of CypA. A follow up of this in the future is warranted.

5. The mechanism whereby T5a restricts poxviruses via L3 binding is not clear, especially since the authors cite that viruses lacking L3 still form intact virions (ref 28). This suggests a different restriction mechanism than disruption of virions by oligomers of T5a binding, as is the case for retroviruses. As well as showing localisation of T5a to viral factories, is it possible to visualise T5a bound to poxviral virions? Is there direct evidence that T5a binds to an intact virion to disrupt it?

We agree with the reviewer that the mechanism of restriction of poxviruses and retroviruses differ in that TRIM5 α is stabilising poxvirus L3 dimers rather than inducing capsid protein degradation as in retroviruses. Additional data addressing this are provided in Fig. S4 and are described in more detail in response to reviewer 2, point 2.

There is no evidence of TRIM5 α binding to intact virions. At post-entry, we did not observe any change in tagged-TRIM5 α localisation that would suggest TRIM5 α recognises intact virions early during infection. Furthermore, there was no co-localisation of TRIM5 α with GFP+ virions in the periphery during egress (see IF images in the above point), which would argue against TRIM5 α binding to intact virions. Additionally, fractionation of the virion showed that L3 is not a membrane-bound protein and therefore, would not be accessible to TRIM5 α when packaged in the intact virion to allow TRIM5 α binding.

6. The confocal images from Fig. S4 would be better placed in the main Figures.

We had placed the confocal images in supplementary only because we could not get the images into the available space in the main figures. If more space is available we would be happy to move these images into the main figures.

7. The section on activation by NFkappaB by T5a and L3 is very under-developed, consisting only of a reporter gene assay analysis of NFkappaB. Activation of innate immunity by T5a in the context of a poxviral infection, and by L3 is an important aspect of the study. It would be quite straightforward for the authors to assess NFkappaB activation with alternative assays, as they have used in their previous papers, in order to gain more insights into how T5a and L3 stimulate NFkappaB gene promoter activation, and to be more convinced of the conclusion (e.g. IkappaB degradation, p65 phosphorylation, p65 nuclear translocation).

To address this issue, we have undertaken two additional assays investigating L3-mediated NF- κ B activation. To do this, we first created a cell line that inducibly expresses L3 and then induced L3 expression by addition of doxycycline for different lengths of time. In parallel, we added doxycyclin to a control cell line containing the empty vector. Cell lysates were blotted for L3, α -tubulin, I κ B α and phospho-TAK-1. The results (Fig. S5c) show that L3 expression induced I κ B α degradation in a time and L3-dependent manner. Although the addition of TNF- α induced the phosphorylation of TAK-1, in the presence of L3 the phospho-TAK-1 levels were not above the control cell line. So we conclude that L3 may activate the NF- κ B pathway somewhere between TAK-1 and I κ B α . As an independent measure of NF- κ B activation, the levels of mRNA of several NF- κ B responsive genes were measured from the same cell lines at 4 h post induction by RT-qPCR. This showed that L3 induced expression of all 5 NF- κ B responsive genes that were tested.

8. Related to (7), how does T5a activate NFkappaB in the context of a poxviral infection? IS it via polyubiquitination of TAK1, as has been shown previously for HIV-1 infection?

TRIM5 α -mediated NF- κ B activation is reported to be at the level of TAK1 autophosphorylation, although in papers reporting this, detection of p-TAK-1 required the addition of bacterial LPS or over-expression of a TRIM5 α -fusion protein. Stimulation was by association with the unanchored polyubiquitin chain TRIM5 synthesises upon its activation by target capsid proteins. It would be logical to hypothesise that this might be via a similar mechanism in the context of poxviral infection. However, as VACV encodes many NF- κ B inhibitors, this makes it technically challenging to assess during infection. Alternatively, we have measured the p-TAK-1 level in L3-expressing cells and found that this was the same as in a control cell line. So we conclude that L3 may activate the pathway somewhere between TAK-1 and I κ B α .

Reviewer Reports on the First Revision:

Referees' comments:

Referee #1 (Remarks to the Author):

This is an important piece of work and there is much to like in the revised manuscript. However, there is perhaps too much here for easy reading. I think it would be very helpful to include a figure comparing what we now know about the mechanisms of Trim5 α action against HIV-1 and poxviruses but this raises the question of what information could be omitted/moved to supplementary figures. I wonder whether there are not in fact too much data here for a "first report" (response P4).

Specific comments

L28. Though more recent work indicates that human Trim5 α may well be able to restrict HIV-1, this is not shown in the reference cited.

L207. S5b

L304. Is "second" correct? The CypA mechanism is surely not directed by VACV.

L331-3. Although CsA and related compounds target a cellular protein might they not select for changes in the dimerization properties of L3?

Referee #2 (Remarks to the Author):

Reviewer #2

In my original review, I praised the novelty and significance of the study. However, the authors have not fully responded to my comments as indicated below.

1. My original comment regarded the question of how L3, a late protein, could manipulate NF- κ B. I think the authors should add a sentence to alert readers to the question, even if proving the mechanism would be beyond the scope of the paper.

2. I agree that the authors have carried out an extensive study. However, my question of whether TRIM5a reduces packaging of L3 resulting in non-infectious virus would seem rather simple to answer.

3. The use of a t-test in an experiment with multiple means is not appropriate. To identify significant differences between specific groups, you need to perform a pairwise comparisons post hoc test following ANOVA. When you use Welch's ANOVA, you can follow with the Games-Howell multiple comparisons method. If you are unfamiliar with this, see:

<https://statisticsbyjim.com/anova/welchs-anova-compared-to-classic-one-way-anova/>

Referee #3 (Remarks to the Author):

My original comments as to the importance and interest of the study stand. And the authors have addressed all the reviewers comments satisfactorily including providing some further mechanistic insights into how T5a restricts poxviruses, and how C6 antagonises this restriction.

Reviewer Reports on the First Revision:

“TRIM5α is a poxvirus restriction factor”

Thank you for your letter of 25th May. We are of course delighted that the paper is accepted in principle subject to the satisfactory revision in light of the points from the reviewers and those in your covering letter. Our detailed response follows.

1. The paper is too long. Please try to shorten it to 8 pages (see the attached proof - this is just a mock proof, not the real thing). To do this you will likely need to move at least 2 main figures to the Extended Data, but as you currently only have 6 ED items (we allow 10) this should be fine. Please also make sure that all figures, including those in the Extended Data are able to fit on a single page. Please eliminate as much white space as possible (especially in the main text figures).

We have revised the figures substantially, including merging figures 5 and 6, and moved many panels into extended data figures, which in consequence have increased to 8 extended data figures. This has saved considerable space by itself, but also enabled the associated figure legends to be simplified and shortened. This has reduced the overall length of the paper by at least 2 full pages.

2. Please provide a revised title within 75 characters (including spaces) that is free of any punctuation marks like colons, exclamation marks, full stops or speech marks. We suggest simply "TRIM5α is a poxvirus restriction factor" (shorter is always more impactful).

The title has been shortened, but we prefer to keep a title that reports both that TRIM5α restricts poxviruses and that these viruses evade this. So we have gone for “TRIM5α restriction of poxviruses and virus evasion strategies” which complies with the length limit.

3. Please provide the manuscript in .docx format.

The manuscript is now provided in .docx format as requested.

4. Methods references should be continuously numbered from the main text methods (but in a separate section, as you have them).

The references have numbered continuously as requested, and the references for the methods remain as a separate list. One issue we had with formatting the lists, is that references that were cited both within the main text and within the references were not listed in the methods section reference section and we hope this is satisfactory.

5. Please remove the main figures from the article file and re-supply them individually in an acceptable format such as EPS, AI, PS, PDF, PPT, CDR, PSD or XLS (for graphs).

The figure files have been removed from the article file and provided as individual files in pdf.

6. Please ensure all main figure legends are as close to 300 words as possible (or fewer).

The figure legends have been simplified and shortened as requested and now are all less than 300 words.

7. Please reduce subheadings to 40 characters (with spaces) or less.

Subheadings have been shorted to comply with the “no more than 40 character” requirement, with one exception, which is 42 characters. Is this OK? In one or two places in the draft typeset article, text had been set as a sub-heading incorrectly and so the removal of these has also reduced space.

8. Please provide a supplementary information guide.

A supplementary information guide has been provided as requested.

9. Please make sure that the data deposited to the iProX repository is available (it is not currently).

The data deposited at iProX has now been made freely available.

10. Please see the attached file for further information about how to revise your paper and Reporting Summary.

We have checked the file for revision of the paper and complied with every request where possible. However, we could not comply with your request 'Please note that the exact p value should be provided, when possible, in the legends of figures 2b, 2f, 2i, 3b, 3g, 3i, 4h, 6d; supplementary figures 2c, 2h, 3a, 5d, 6c, 6f.', This is because these p values are less than 0.0001 and most have more than 15 0's after the decimal point, which is beyond the limit of what GraphPad Prism can give, and so these have been left as $p < 0.0001$. Other than this, the editorial requests have been dealt with.

TRANSPARENT PEER REVIEW: *'We do wish to participate in transparent peer review'*

Referees' comments:

Referee #1 (Remarks to the Author):

This is an important piece of work and there is much to like in the revised manuscript. However, there is perhaps too much here for easy reading. I think it would be very helpful to include a figure comparing what we now know about the mechanisms of Trim5 α action against HIV-1 and poxviruses but this raises the question of what information could be omitted/moved to supplementary figures. I wonder whether there are not in fact too much data here for a “first report” (response P4).

We thank the reviewer for the thoughtful comment. As indicated by the Editor, it is necessary that we retain all the data provided in the revised manuscript, but we have re-organised these data and moved a lot of material to the extended data figures. We believe this gives a more streamline paper for readers, whilst retaining all the data that support the conclusions drawn.

Specific comments

L28. Though more recent work indicates that human Trim5 α may well be able to restrict HIV-1, this is not shown in the reference cited.

Additional references in support of this statement have been added. This now takes the total to 51 (rather than 50) and we hope this is acceptable.

L207. S5b

We thank the reviewer for spotting this mistake. The re-organisation and condensation of the figures has meant that there has been substantial re-numbering and re-lettering of the data panels.

L304. Is “second” correct? The CypA mechanism is surely not directed by VACV.

We may have a different interpretation to the reviewer here. The L3 protein has evolved a structure that is able to interact with CypA and thereby recruit it into virions, and this interaction is beneficial to the virus. It is therefore likely that there has been positive selection for L3 to do this. So we think it reasonable to describe this as a second mechanism by which the virus has evolved to escape from the restriction imposed by TRIM5a.

L331-3. Although CsA and related compounds target a cellular protein might they not select for changes in the dimerization properties of L3?

It is possible that the use of CsA or derivatives might eventually lead to changes in VACV L3 so L3 can now bind a cyclophilin A / CsA complex. However, this would still require the complex to retain prolyl isomerase activity (which is needed for the proviral activity) and given that currently the interaction of L3 and CypA is interrupted by CsA, it would likely require a new interaction surface. Collectively, this would seem rather unlikely, although we agree that one can never say never. We feel that the tone of the current draft (“making the emergence of drug resistance difficult”) captures unlikely scenario while not ruling it out entirely.

Referee #2 (Remarks to the Author):

Reviewer #2

In my original review, I praised the novelty and significance of the study. However, the authors have not fully responded to my comments as indicated below.

1. My original comment regarded the question of how L3, a late protein, could manipulate NF- κ B. I think the authors should add a sentence to alert readers to the question, even if proving the mechanism would be beyond the scope of the paper.

We agree that determining how L3 induces activation of NF- κ B is beyond the scope of the current manuscript. As requested, we have included a sentence in the revised paper to address this issue. “L3 is expressed late during infection, after NF- κ B activation is suppressed by many viral inhibitors expressed early during infection, but L3 might be detected during uncoating of incoming virions unless encapsidated CypA blocks this.”

2. I agree that the authors have carried out an extensive study. However, my question of

whether TRIM5a reduces packaging of L3 resulting in non-infectious virus would seem rather simple to answer.

We agree that this is a simple (and very sensible) question to ask, but might disagree that this is easy to answer. The difference in infectious virus titre of 2-3 fold is relatively small so that many of the virions produced in the presence of TRIM5a are infectious and many human cell lines that have been used routinely to propagate VACV are TRIM5a positive. Given that TRIM5a enhances L3 dimerisation, the defect might not be in packaging but could be due to biochemical modification of L3. After discussing this issue raised by the referee with the Senior Editor it was agreed that this question need not be addressed in this first report, but will be addressed in a follow up study.

3. The use of a t-test in an experiment with multiple means is not appropriate. To identify significant differences between specific groups, you need to perform a pairwise comparisons post hoc test following ANOVA. When you use Welch's ANOVA, you can follow with the Games-Howell multiple comparisons method. If you are unfamiliar with this, see:

<https://statisticsbyjim.com/anova/welchs-anova-compared-to-classic-one-way-anova/>

We are grateful to the reviewer for the additional advice on statistical analysis, and have undertaken the analysis with a similar method that provides specific p values for the individual groups in experiments with multiple means.

Referee #3 (Remarks to the Author):

My original comments as to the importance and interest of the study stand. And the authors have addressed all the reviewers comments satisfactorily including providing some further mechanistic insights into how T5a restricts poxviruses, and how C6 antagonises this restriction.

We are grateful to the reviewer for the positive summary and recommendation.

We hope that the additional data provided, textual changes and explanations to address the issues raised are satisfactory.